# Phosphatase PTPN22 functions as an adaptor in the mTORC2 complex

Keshav Gupta [ID] [1,2], Nagalakshmi Kommineni[1], Tanuja Bogadi[1], Neeraja P Alamuru-Yellapragada[1] & Subbareddy Maddika [ID] [1✉]

## Abstract

mTOR (mechanistic target of rapamycin) kinase is a pivotal regulator of cellular growth and metabolism, integrating signals from nutrients and growth factors. It functions through the assembly of two distinct complexes, mTORC1 and mTORC2, which differ in their substrate specificity and regulation. While the regulation of mTORC1 is well-characterized, less is known about the modulators of mTORC2 signaling. In this study, we identify tyrosine phosphatase PTPN22 as an mTORC2-associated protein. We provide evidence that PTPN22 is essential for the activation of the mTORC2/AKT axis, independent of cell lineage. Loss of PTPN22 results in impaired AKT phosphorylation in response to both basal and growth factor signals. Mechanistically, PTPN22 functions as a scaffolding protein that promotes the mSIN-RICTOR interaction, thereby maintaining mTORC2 complex integrity. Notably, this adaptor function of PTPN22 is independent of its tyrosine phosphatase activity. Functionally, we demonstrate that PTPN22 is required for cell growth and survival in both cellular models and nude mouse xenografts. Together, these findings reveal a non-catalytic role for phosphatase PTPN22 in mTORC2 assembly and function.

**Keywords** mTOR; PTPN22; AKT; Rictor; mSIN
**Subject Category** Signal Transduction

## Introduction

The mechanistic target of rapamycin (mTOR) is a serine/threonine kinase that exists as the catalytic core of two distinct multi-subunit complexes, mTORC1 and mTORC2, which play essential roles in regulating cellular growth, metabolism, and survival in response to growth factors and nutrient availability (Kim and Guan, 2019; Sabatini, 2017). mTORC1, composed of mTOR, RAPTOR (Regulatory associated protein of mTOR), mLST8 (mammalian lethal with SEC13 protein 8), and its inhibitors PRAS40 (Proline-rich Akt substrate of 40 kDa) and DEPTOR (DEP domain containing mTOR interacting protein), is primarily activated by nutrient signals such as amino acids, and translocates to the lysosomal membrane (Liu and Sabatini, 2020; Saxton and Sabatini, 2017). Upon activation, mTORC1 phosphorylates a range of substrates to regulate cell growth and metabolism (Garami et al, 2003; Sancak et al, 2008; Saxton and Sabatini, 2017). Similarly, mTORC2, composed of mTOR, mLST8, DEPTOR, RICTOR (Rapamycin-insensitive companion of mTOR), and mSIN1 (mammalian stress-activated protein kinase interacting protein), is primarily activated by growth factors through the classical PI3K signaling pathway. This activation is mediated by the binding of phosphatidylinositol (3,4,5)-trisphosphate (Ptdlns(3,4,5)$P_3$) to the Pleckstrin homology (PH) domain of mSIN1, which relieves its autoinhibition and enables mTORC2 to phosphorylate and activate AKT (Kim and Guan, 2019; Liu and Sabatini, 2020; Saxton and Sabatini, 2017). mTORC2 promotes cell proliferation and survival by phosphorylating AKT at serine 473 residue (Sarbassov et al, 2006; Sarbassov et al, 2005). Moreover, mTORC2 also phosphorylates other AGC family of protein kinases including SGK1, PKCα, and PKCϒ to regulate ion transport, cytoskeleton remodelling, and cell migration (Baffi et al, 2021; Garcia-Martinez and Alessi, 2008; Ikenoue et al, 2008; Liu and Sabatini, 2020; Sarbassov et al, 2004). In response to nutrients and growth factors availability, multiple proteins interact with mTOR complexes to modulate their functions (Bracho-Valdes et al, 2011). Although the regulatory mechanisms of mTORC1 are well-characterized, less is known about the factors that modulate mTORC2. In this study, we used interactome mapping, biochemical, genetic, and functional analyses to identify PTPN22 (protein tyrosine phosphatase non-receptor type 22) as a key regulator of mTORC2 activity and function.

PTPN22 consists of a highly conserved catalytic domain with intrinsic tyrosine phosphatase activity and belongs to the family of non-receptor protein tyrosine phosphatases (Cohen et al, 1999; Jassim et al, 2022; Matthews et al, 1992; Stanford and Bottini, 2014). PTPN22 is highly expressed in immune cells including T cells, B cells and dendritic cells (Bottini and Peterson, 2014). PTPN22 suppress T cell activation by dephosphorylating several critical modulators of T cell receptor signaling including Lck, ZAP70, TCRζ and VAV1 (Cloutier and Veillette, 1999; Gjorloff-Wingren et al, 1999; Hasegawa et al, 2004; Stanford and Bottini, 2014; Vang et al, 2012; Wu et al, 2006). Moreover, by regulating other immune cells functions it modulates several aspects of innate

[1]Laboratory of Cell Death & Cell Survival, Centre for DNA Fingerprinting and Diagnostics (CDFD), Uppal, Hyderabad, Telangana 500039, India. [2]Graduate Studies, Manipal Academy of Higher Education, Manipal, Karnataka 576104, India. ✉E-mail: msreddy@cdfd.org.in

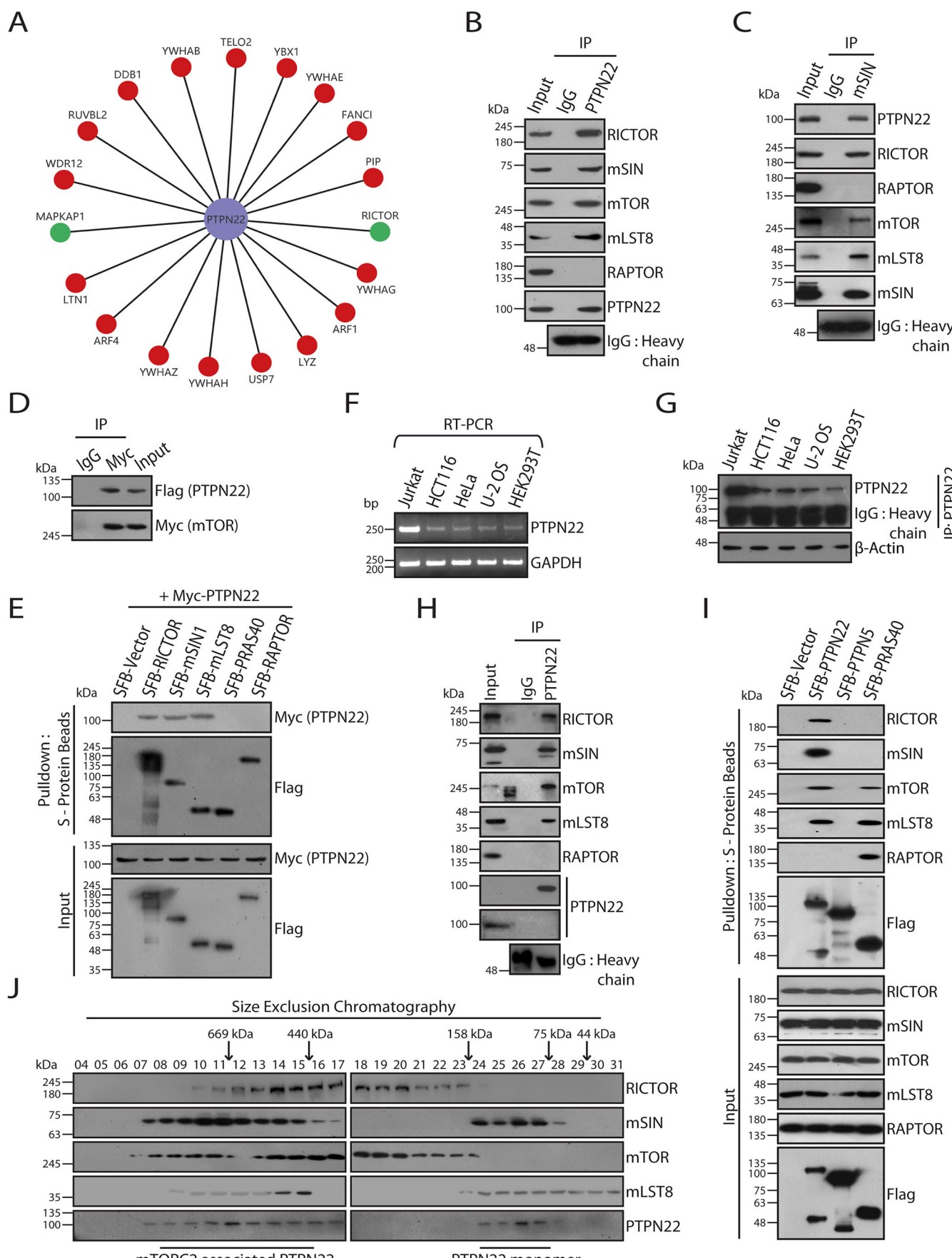

**Figure 1. PTPN22 associates with intact mTORC2 kinase but not with mTORC1.**

(A) Partial interaction network of PTPN22 with its associated proteins derived from tandem affinity purification of SFB-PTPN22 followed by peptides identification by mass spectrometry analysis is shown. SFB-GFP purification was used as a control to filter out non-specific interactions. The list of SFB-PTPN22 and SFB-GFP associated proteins was derived from our earlier study (Kumar et al, 2017). mTOR complex components are highlighted in green colour. (B) Immunoprecipitation (IP) with control IgG or anti-PTPN22 antibody was performed with extracts derived from Jurkat cells lysed in 0.3% CHAPS buffer. Endogenous association of PTPN22 with mTOR complexes subunits (RICTOR, mSIN, mTOR, mLST8, and RAPTOR) were analysed by immunoblotting, immunoprecipitates with respective antibodies. (C) Jurkat cells were lysed in 0.3% CHAPS buffer and immunoprecipitates from control IgG or anti-mSIN antibody were analysed for the presence of PTPN22, and mTOR components by immunoblotting with their respective antibodies. (D) HEK293T cells were co-transfected with SFB-PTPN22 and Myc-mTOR constructs. At 24 h post-transfection, cells were lysed in 0.3% CHAPS buffer, followed by immunoprecipitation (IP) with control IgG or anti-Myc antibody and their interaction was detected by immunoblotting with anti-Flag antibody. (E) SFB vector, SFB-RICTOR, SFB-mSIN1, SFB-mLST8, SFB-PRAS40 or SFB-RAPTOR along with Myc-PTPN22 were transfected in HEK293T cells. At 24 h post-transfection, cells were lysed in 0.3% CHAPS buffer and were subjected to pulldown using S-protein agarose beads. Association of PTPN22 with components of mTORC1 and mTORC2 were analysed by immunoblotting with anti-Myc antibody. (F) Detection of PTPN22 expression by RT-PCR in different cell lines. GAPDH was used as a control. (G) Immunoprecipitation (IP) with anti-PTPN22 antibody was performed with extracts derived from indicated cell lines lysed in 0.5% NETN buffer. Endogenous levels of PTPN22 were analysed by immunoblotting, immunoprecipitates with anti-PTPN22 antibody. β-actin was used as a control. (H) HEK293T cells were lysed in 0.3% CHAPS buffer and immunoprecipitates from control IgG or anti-PTPN22 antibody were analysed for the presence of components of mTOR complexes by immunoblotting with their respective antibodies. As levels of endogenous PTPN22 are low in HEK293T cells, expression of PTPN22 in input sample was shown by immunoprecipitating PTPN22 from cell extracts using its antibody. (I) HCT116 cells were transfected with either SFB vector, SFB-PTPN22, SFB-PTPN5 or SFB-PRAS40. At 48 h post-transfection, cells were lysed in 0.3% CHAPS buffer and were subjected to pulldown using S-protein agarose beads. Interaction with subunits of mTOR complexes were detected by immunoblotting with their respective antibodies. (J) Jurkat cells were lysed in 0.3% CHAPS buffer and were fractionated using FPLC Superdex 200 column (gel-filtration chromatography). Molecular weights for fractions were estimated by running native molecular weight markers. Indicated fractions were resolved on SDS-PAGE followed by immunoblotting with respective antibodies as mentioned. Source data are available online for this figure.

and adaptive immune signaling including B cell antigen receptor (BCR) signaling, dectin-1 signaling in dendritic cells, NLRP3-mediated and NOD2-mediated signaling in macrophages, toll-like receptor-mediated signaling in myeloid cells and IgE-mediated signaling in mast cells (Menard et al, 2011; Obiri et al, 2012; Purvis et al, 2018; Rieck et al, 2007; Spalinger et al, 2017; Spalinger et al, 2013; Wang et al, 2013). Clinically, a single missense PTPN22 variant (R620W) is strongly associated with an increased risk of multiple autoimmune disorders, including rheumatoid arthritis (Begovich et al, 2004), type 1 diabetes mellitus (Bottini et al, 2004), and systemic lupus erythematosus (Kyogoku et al, 2004).

Although PTPN22 is well studied in the context of immune regulation, its function outside of the immune system remains poorly understood. It has been detected in non-hematopoietic tissues, and in animal models lacking a functional adaptive immune system, but its role in these systems is not well defined (Binti et al, 2024; Karlsson et al, 2021; Sjostedt et al, 2020; Thul et al, 2017; Uhlen et al, 2015; Xue et al, 2022). Here, we present evidence that PTPN22 is a novel regulator of mTORC2 assembly and function, expanding our understanding of its biological roles beyond the immune system.

## Results and discussion

### PTPN22 associates with mTORC2 complex

In an attempt to gain insights into new cellular functions of PTPN22, we analyzed the interactome of PTPN22 mapped through tandem affinity purification coupled with mass spectrometry analysis in our earlier study (Kumar et al, 2017). Interestingly, pathway enrichment analysis of PTPN22 interacting proteins using Enrichr tool (Kuleshov et al, 2016) revealed enrichment of proteins associated with multiple cellular pathways including c-Myc transcriptional activation, ErbB1 downstream signaling and PI3K/AKT/mTOR signaling (Fig. EV1A). Among these, we repeatedly found RICTOR and MAPKAP1 (also known as mSIN1), the mTORC2-specific components in PTPN22 purified complex,

suggesting a potential new role of PTPN22 in the mTOR pathway (Fig. 1A). Firstly, we tested the endogenous association of PTPN22 with mTORC1 and mTORC2 subunits in Jurkat T cells. Immunoprecipitation under mild lysis conditions (0.3% CHAPS buffer), which preserve the integrity of mTOR complexes (Kim et al, 2002; Pearce et al, 2007), demonstrated that PTPN22 co-immunoprecipitated with RICTOR, mSIN (mTORC2-specific subunits), mTOR, and mLST8 (common to both mTORC1/C2), but not RAPTOR (mTORC1-specific subunit) (Fig. 1B). These data suggest a specific interaction between PTPN22 and mTORC2. Reciprocally, endogenous mSIN associates with PTPN22 along with other subunits of mTORC2 (Fig. 1C), suggesting PTPN22-mTORC2 interaction is physiologically relevant in cellular milieu. To further confirm this interaction, we conducted co-immunoprecipitation experiments in which we overexpressed PTPN22 alongside mTORC1 and mTORC2 components. mTOR, RICTOR, mSIN, and mLST8 co-immunoprecipitated with PTPN22, while RAPTOR and PRAS40 did not (Fig. 1D,E). Reverse co-immunoprecipitation using PTPN22 as bait also selectively pulled down RICTOR, mSIN, mTOR, and mLST8, confirming the specificity of the interaction with mTORC2 (Fig. EV1B). These findings indicate that PTPN22 specifically associates with mTORC2 complex.

While PTPN22 is known to be expressed in hematopoietic cells and involved in immune signaling (Bottini and Peterson, 2014; Cloutier and Veillette, 1996; Cohen et al, 1999; Matthews et al, 1992; Stanford and Bottini, 2014; Zhang et al, 2020), its expression and role in non-hematopoietic cells remain less explored. To address this, we screened epithelial cell lines (HCT116, HeLa, HEK293T, and U-2OS) for PTPN22 expression using RT-PCR. As a positive control, Jurkat T cells, known for their high expression of PTPN22, were included. We found detectable levels of PTPN22 expression in all epithelial cell lines tested (Fig. 1F). Additionally, we detected PTPN22 expression at protein level in the same panel of cell lines by immunoprecipitating PTPN22 from cell extracts by using its antibody (Fig. 1G). Next, we confirmed the association of endogenous PTPN22, as well as transiently expressing exogenous SFB-PTPN22 with components of mTORC2 in multiple epithelial

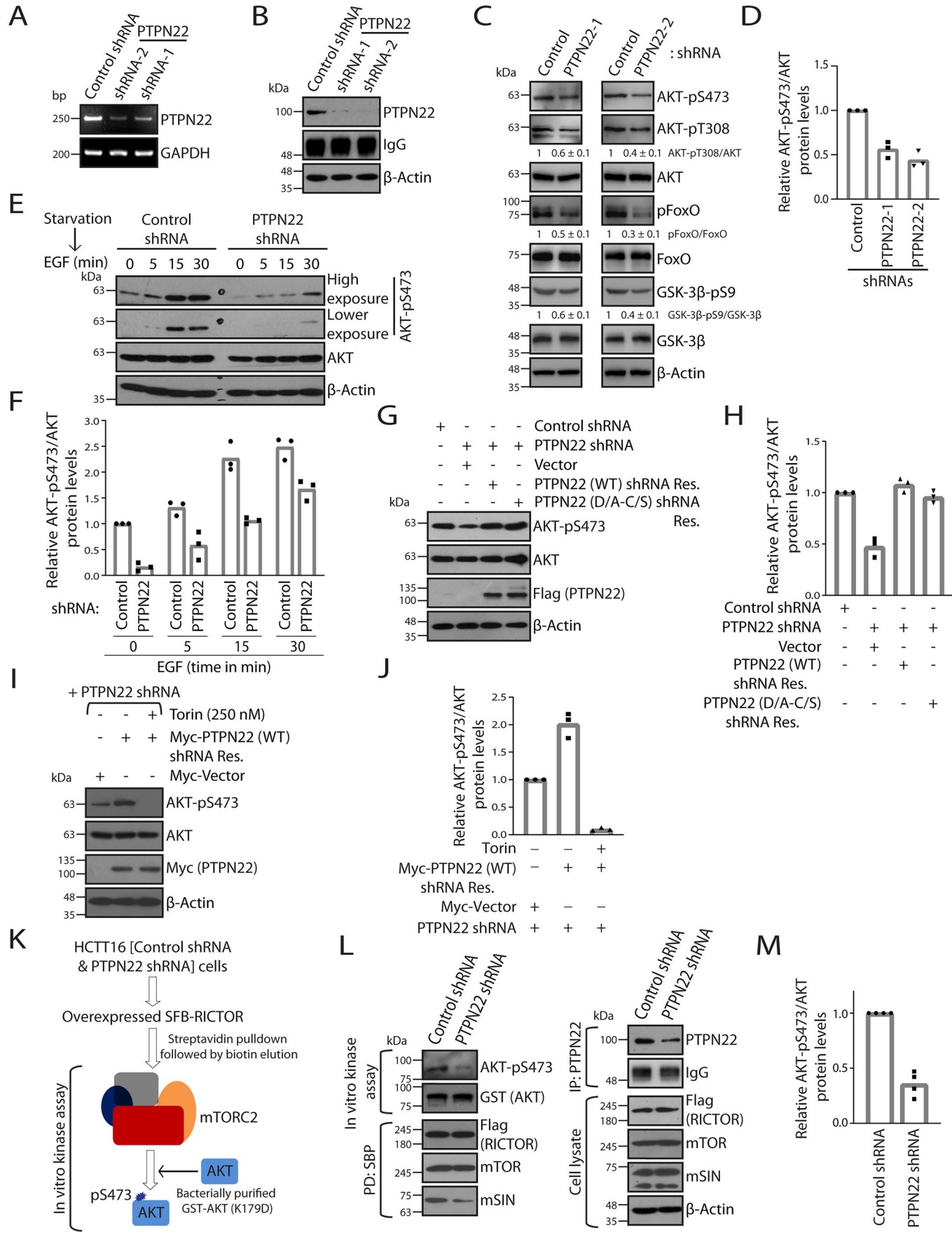

**Figure 2.  PTPN22 promotes mTORC2 activation independent of its catalytic activity.**

(A, B) HCT116 cells were transduced with either control or two independent PTPN22 shRNAs, and stable cell lines were generated. Knockdown of PTPN22 was confirmed either by (A) examining the expression levels of PTPN22 and GAPDH mRNAs by reverse transcription polymerase chain reaction (RT-PCR), (B) or by immunoprecipitating PTPN22 from cell lysates using endogenous antibody, followed by immunoblot analysis of whole cell lysates and anti-PTPN22 immunoprecipitates with antibodies as indicated. (C, D) Immunoblot analysis of whole cell lysates derived from stable HCT116 cell lines expressing the control shRNA or two independent PTPN22 shRNAs, with indicated antibodies was done to determine the activation of AKT and its downstream substrates (C). Quantification of immunoblotting data, the pixel intensity of AKT-pS473 protein bands were normalized by the pixel intensity of total AKT protein bands. Individual data points from three independent experiments are shown in the graph (D), and the phospho-protein/total protein levels of pAKT-T308, pGSK-3β-pS9 and pFoxO are indicated as mean ± SD from three independent experiments are shown below their respective blots. (E, F) Control or PTPN22 depleted HCT116 cells were serum starved for 16 h, followed by stimulation with EGF (100 ng/ml) at indicated time points. Whole cell lysates were immunoblotted with indicated antibodies to determine the activation of AKT (E). Quantification of immunoblotting data, the pixel intensity of AKT-pS473 protein bands were normalized by the pixel intensity of total AKT protein bands. Individual data points from three independent experiments are shown in the graph (F). (G, H) Whole cell lysates of control shRNA or PTPN22 shRNA transfected with SFB-vector, SFB-PTPN22 (WT) or SFB-PTPN22 (D/A-C/S) shRNA-resistant plasmids were examined to determine the total and phosphorylated states of AKT with specific antibodies (G). Quantification of immunoblotting data, the pixel intensity of AKT-pS473 protein bands were normalized by the pixel intensity of total AKT protein bands. Individual data points from three independent experiments are shown in the graph (H). (I, J) PTPN22 depleted HCT116 cells expressing Myc-PTPN22 (WT) shRNA-resistant construct were treated with torin (250 nM) for 2 h. Cell lysates were analysed for total and phosphorylated states of AKT with specific antibodies. Untreated cells or PTPN22 depleted HCT116 cells expressing Myc-vector were used as controls (I). Quantification of immunoblotting data, the pixel intensity of AKT-pS473 protein bands were normalized by the pixel intensity of total AKT protein bands. Individual data points from three independent experiments are shown in (J). (K) Schematic showing experimental workflow of in vitro mTORC2 kinase assays using bacterially purified GST-AKT (K179D) as a substrate. (L, M) mTORC2 purified from control or PTPN22 depleted HCT116 cells were incubated with bacterially purified GST-AKT (K179D) for in vitro kinase reactions, and substrate phosphorylation was assessed with AKT-pS473 antibody. Cell lysates and pulldowns were immunoblotted with the indicated antibodies (L). Quantification of immunoblotting data, the pixel intensity of AKT-pS473 protein bands were normalized by the pixel intensity of total AKT protein bands. Individual data points from three independent experiments are shown in the graph (M). Source data are available online for this figure.

cell lines (Figs. 1H,I and EV1C). To assess whether the interaction of PTPN22 with mTORC2 is specific to this phosphatase, we performed a co-IP with PTPN5, another tyrosine phosphatase. Unlike PTPN22, PTPN5 did not co-immunoprecipitate with mTORC2 (Fig. 1I), suggesting that PTPN22's interaction with mTORC2 is specific. Consistent with these findings, PTPN22 co-fractionate along with mTORC2 complex in our size fractionation experiments (Fig. 1J). Together, our data indicate that PTPN22 interacts with mTORC2 in a wide range of cells derived from haematopoietic as well as non-haematopoietic origin.

## PTPN22 activates mTORC2-AKT axis independent of its catalytic activity

Given PTPN22's association with mTORC2, we next investigated whether it modulates the activity of mTORC2. We analyzed the phosphorylation of AKT (a bonafide substrate of mTORC2) as a measure of activity of functional mTORC2 (Sarbassov et al, 2005). Lentiviral shRNA-mediated knockdown of PTPN22 in Jurkat cells resulted in reduced phosphorylation of AKT at Ser473 and Thr308, without affecting mTORC1 activity (p70S6K phosphorylation at Thr389) (Fig. EV2A–D). This was accompanied by decreased phosphorylation of downstream AKT substrates, GSK-3β and FoxO1 (Fig. EV2A,E,F). Further, we generated stable PTPN22 knockdown HCT116 cells (Fig. 2A,B) and observed a similar reduction in AKT phosphorylation and its downstream targets (Fig. 2C,D), suggesting PTPN22 effect on mTORC2 activation is independent of cell-type. Moreover, we assessed the phosphorylation status of other mTORC2 targets, SGK1 and PKCα. Notably, we found that, the phosphorylation of SGK1-pS422 and phospho-PKCα/βII [pT638/641] were decreased in PTPN22 depleted cells in comparison to control cells (Fig. EV2G–I). In line with these findings, PTPN22 knockout HCT116 cells also displayed reduced phosphorylation of AKT-pS473 and GSK-3β-pS9 as compared to control cells (Fig. EV2J). Additionally, knockdown of PTPN22 impaired growth factor-induced AKT activation (Fig. 2E,F), suggesting a role for PTPN22 in mTORC2-AKT signaling under both basal and growth factor-stimulated conditions. Given that PTPN22 is a phosphatase, we next tested the role of its catalytic activity in AKT activation by exogenously expressing wild-type or catalytically inactive mutant of PTPN22 (D195A/C227S) in PTPN22 depleted background. Surprisingly, the expression of phosphatase dead form of PTPN22 could restore the defect in the AKT-pS473 phosphorylation in PTPN22 depleted cells similar to wild-type (Fig. 2G,H), possibly suggesting a phosphatase activity-independent role of PTPN22 in AKT activation. On the other hand, ectopic expression of either wild-type or catalytically dead PTPN22 resulted in increased phosphorylation of AKT (Ser473) and its substrate FoxO (Fig. EV2K). Moreover, the enhanced AKT-pS473 phosphorylation in PTPN22 expressing cells is strongly reduced upon treatment with Torin1, a potent mTOR inhibitor (Fig. 2I,J), suggesting that the regulation of AKT by PTPN22 is through mTORC2. To further test if this is indeed true, we performed an in vitro mTORC2 kinase assay, where we affinity purified SFB-RICTOR from HCT116 cells stably expressing control shRNA or PTPN22 shRNA, and used as a kinase source to assess their ability to phosphorylate bacterially purified recombinant AKT (K179D) (Fig. 2K). In line with our hypothesis, knockdown of PTPN22 suppressed mTORC2's ability to phosphorylate recombinant AKT in vitro, as assessed by serine 473 phosphorylation on AKT (Fig. 2L,M). Collectively, our results demonstrate a possible non-catalytic role of PTPN22 in modulating mTORC2 kinase activity in cells as well as in vitro.

## PTPN22 promotes mTORC2 assembly by facilitating mSIN and RICTOR association

Next, we explored the mechanism on how PTPN22 enhances mTORC2 activity in cells. The initial clue about the possible mechanism of mTORC2 regulation by PTPN22 came from mTORC2 kinase preparation during our in vitro kinase assays, where we observed reduced binding of mSIN with RICTOR in PTPN22 knockdown cells in comparison with control cells (Fig. 2L,M). This suggested a potential adaptor role of PTPN22

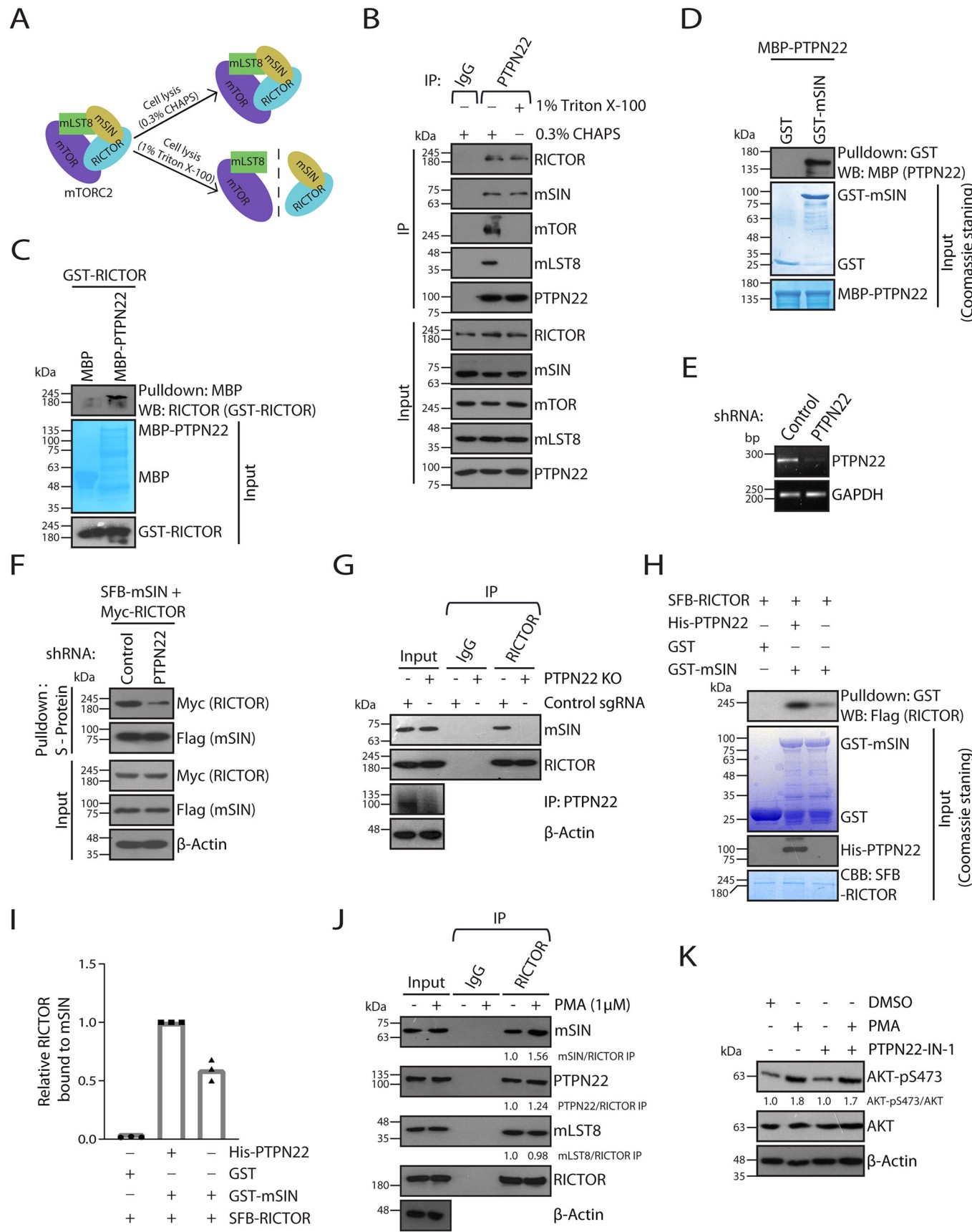

Figure 3. PTPN22 promotes mTORC2 assembly by mediating RICTOR-mSIN association.

(A) A schematic showing sensitivity of mTORC2 towards different detergents during isolation of mTORC2. Cell lysis with 0.3% CHAPS buffer preserves mTORC2 integrity, while 1% Triton X-100 containing buffer cause dissociation of mTORC2 into two modules; one having mTOR and mLST8, and other having RICTOR and mSIN. (B) Immunoprecipitation (IP) with control IgG or anti-PTPN22 antibody was performed with extracts derived from Jurkat cells lysed in a buffer containing either 0.3% CHAPS or 1% Triton X-100. Endogenous association of PTPN22 with mTORC2 components were analysed by immunoblotting with respective antibodies. (C) Bacterially expressed recombinant MBP or MBP-PTPN22 proteins immobilized to dextran Sepharose beads were incubated with concentrated bacterial cell lysates expressing GST-RICTOR. The MBP-pulldowns were resolved by SDS-PAGE and the interactions were analyzed by immunoblotting with anti-RICTOR antibody. Expression of MBP and MBP-PTPN22 was shown by Coomassie staining. (D) Bacterially expressed recombinant GST, GST-mSIN and MBP-PTPN22 were purified using glutathione sepharose and dextran sepharose beads, respectively. 2 µg of purified MBP-PTPN22 was incubated with glutathione sepharose beads bound GST or GST-mSIN. The GST-pulldowns were resolved on SDS-PAGE and analysed by immunoblotting with anti-MBP antibody to check for the interaction. Recombinant protein expression was shown by Coomassie staining. (E) Lentiviral transduction with control shRNA or PTPN22 shRNA was performed in HCT116 cells. Knockdown of PTPN22 was verified by examining the expression levels of PTPN22 and GAPDH mRNAs by reverse transcription polymerase chain reaction (RT-PCR). (F) Control or PTPN22 depleted HCT116 cells were co-transfected with SFB-mSIN and Myc-RICTOR. At 48 h post-transfection, cells were lysed in 0.3% CHAPS buffer and lysates were pulldown using S-protein agarose beads. The interactions were detected by immunoblotting with anti-Myc antibody. (G) Immunoprecipitation (IP) with control IgG or anti-RICTOR antibody was performed with extracts derived from either control sgRNA or PTPN22 knockout HCT116 cells. Endogenous association of RICTOR with mSIN were analysed by immunoblotting with anti-mSIN antibodies. Due to low expression level of PTPN22 in HCT116 cells, endogenous PTPN22 in input sample was shown by immunoprecipitating PTPN22 from cell extracts using its antibody. (H, I) Bacterially purified GST-mSIN immobilized on glutathione sepharose beads were incubated with the purified SFB-RICTOR, either in the presence of recombinant PTPN22 or equal volume of corresponding buffer. The interaction of mSIN-RICTOR was assessed by immunoblotting with anti-Flag antibody. GST-protein was used as a negative control (H). Individual data points for relative RICTOR bound to mSIN were plotted as graph from three independent experiments (I). (J) Jurkat cells treated either with phorbol 12-myristate 13-acetate (PMA, 1 µM) or Dimethylsulphoxide (DMSO) were lysed in 0.3% CHAPS buffer and immunoprecipitates from control IgG or anti-RICTOR antibody were analysed for the presence of PTPN22, and mTOR components by immunoblotting with their respective antibodies. (K) Immunoblot (IB) analysis of whole cell lysates derived from Jurkat cells treated with DMSO, PMA (1 µM), PTPN22-IN-1 (1.4 µM) or PMA treatment followed by PTPN22-IN-1 treatment, with indicated antibodies to determine the activation of AKT. Source data are available online for this figure.

in mTORC2 assembly via protein-protein interaction. Previous studies have reasonably utilized differential detergent cell lysis in order to identify the specific mTOR subunit(s) that directly interacts with a protein in question (Kaizuka et al, 2010; Kim et al, 2002; Kim et al, 2003; Pearce et al, 2007; Sarbassov et al, 2004; Zhang et al, 2016). Cell lysis with 1% (v/v) Triton X-100 containing buffer disrupts mTORC2 into two modules: mTOR-mLST8 and RICTOR-mSIN, whereas, lysis with 0.3% (w/v) CHAPS buffer helps to preserve the integrity of mTOR complexes (Fig. 3A). Therefore, in order to understand how PTPN22 activates mTORC2, we sought to identify the subunits of mTORC2 that directly binds with PTPN22. Although PTPN22 co-immunoprecipitated with the intact mTORC2 complex in CHAPS buffer, we observed that mSIN and RICTOR, but not mTOR and mLST8, associated with PTPN22 in Triton X-100 buffer (Fig. 3B). These observations were consistent with our mass spectrometry results, where we observed the enrichment of only mSIN and RICTOR but not of mTOR and mLST8 in PTPN22 purification, as we employed NP-40 (Nonidet P-40) detergent-based buffer for cell lysis, which has similar effects on mTOR complexes integrity as that of Triton X-100 containing buffer (Fig. 1A). More importantly, bacterially purified recombinant MBP-PTPN22 can associate with bacterially expressing recombinant GST-RICTOR (Fig. 3C). In addition, bacterially purified GST-tagged mSIN but not GST interacted with recombinant MBP-PTPN22 (Fig. 3D), possibly suggesting a direct interaction of PTPN22 with RICTOR and mSIN. To substantiate the potential adaptor role of PTPN22 in mTORC2 assembly, we analysed the interaction of mTORC2 subunits in PTPN22 depleted cells. Notably, depletion of PTPN22 (Fig. 3E) markedly diminished mSIN-RICTOR association (Fig. 3F) without altering the binding of mSIN-mLST8 (Fig. EV3A), RICTOR-mTOR (Fig. EV3B), and mTOR-mLST8 (Fig. EV3C). PTPN22 knockout HCT116 cells also show diminished mSIN-RICTOR association (Fig. 3G). Furthermore, our in vitro binding studies suggested that PTPN22 enhances mSIN-RICTOR association (Fig. 3H,I). These results suggest a potential scaffolding role of PTPN22 in mTORC2 assembly via its

interaction with mSIN and RICTOR. Next, to test if growth factors affect PTPN22-mTORC2 interaction, we performed co-immunoprecipitation of SFB-PTPN22 with RICTOR and mSIN, with or without addition of growth factor (EGF). Interaction of PTPN22 with mSIN and RICTOR remains unchanged (Fig. EV3D) with addition of EGF. Importantly, in HCT116 cells, depletion of PTPN22 did not maintain RICTOR-mSIN association under both conditions, serum-starvation or in response to epidermal growth factor (EGF) (Fig. EV3E), suggesting PTPN22 is required for RICTOR-mSIN association, irrespective of the availability of growth factors. Further to substantiate the potential role of PTPN22 protein in promoting RICTOR-mSIN association, we treated Jurkat cells with phorbol 12-myristate 13-acetate (PMA), which is reported to cause elevated levels of PTPN22 protein (Negro et al, 2012; Yang et al, 2020). In line with our findings, treatment with PMA led to an increase in binding between mSIN and RICTOR (Fig. 3J). In addition to this, we demonstrated that the treatment with PMA promotes the phosphorylation of AKT. However, this increased phosphorylation of AKT is not due to PTPN22 phosphatase activity, as pharmacological inactivation of PTPN22 activity by treating cells with the PTPN22 inhibitor (PTPN22-IN-1) does not affect AKT phosphorylation. Moreover, treatment with PTPN22-IN-1 is not able to rescue the increased AKT phosphorylation by PMA (Fig. 3K). Taken together, the PTPN22 protein levels, but not the activity is required for RICTOR-mSIN interaction and proper mTORC2 activation.

PTPN22 protein has an N-terminal PTP (protein tyrosine phosphatase) catalytic domain and a C-terminal domain with four proline-rich motifs that mediate interaction with other proteins, connected by a central interdomain having regulatory function (Bottini and Peterson, 2014; Jassim et al, 2022; Stanford and Bottini, 2014). To further understand the molecular details of mSIN-RICTOR and PTPN22 interacting regions, we generated structural models using Alphafold. The predicted structural model of PTPN22, mSIN1 and RICTOR displayed an interaction of mSIN with the N-terminal region of PTPN22, while the C-terminal

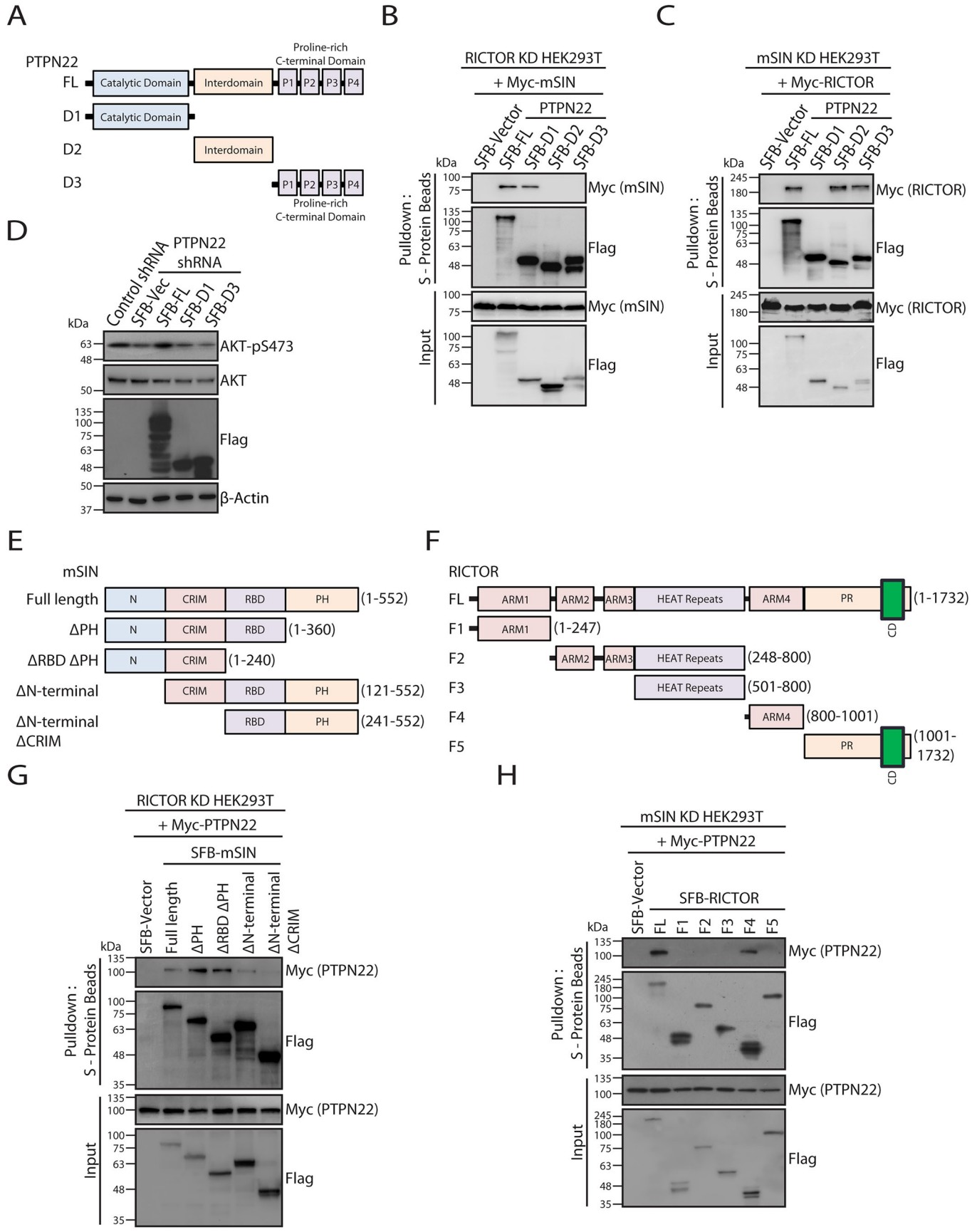

Figure 4. Mapping interaction domains essential for mediating the PTPN22-mSIN and PTPN22-RICTOR binding.

(A) A schematic showing the full length PTPN22 along with its various deletion mutants lacking indicated domains. (B) SFB vector, SFB-PTPN22 full length (FL) or SFB-tagged deletion constructs of PTPN22 (shown in (A)) along with Myc-mSIN were co-transfected in HEK293T cells having stable knockdown (KD) of RICTOR. At 24 h post-transfection, cells were lysed in 1% Triton X-100 buffer and were subjected to pulldown using S-protein agarose beads, and their interactions were analysed by immunoblotting with anti-Myc antibody. (C) SFB vector, SFB-PTPN22 full length (FL) or SFB-tagged deletion constructs of PTPN22 (shown in (A)) along with Myc-tagged RICTOR were co-transfected in HEK293T cells having stable knockdown (KD) of mSIN. 24 h after transfection, cells were lysed in 1% Triton X-100 buffer and lysates were pulldown with S-protein agarose beads, and their interactions were analysed by immunoblotting with anti-Myc antibody. (D) HCT116 cells stably expressing PTPN22 shRNA were transfected either with SFB-vector, SFB-PTPN22 (WT) shRNA-resistant plasmids or SFB-tagged D1 and D3 deletion constructs of PTPN22. Whole cell lysates were immunoblotted to determine the activation of AKT by examined the total and phosphorylated states of AKT with specific antibodies. HCT116 cells expressing scrambled shRNA were used as a control. (E) Schematic representation of mSIN full length, and its various truncation mutants lacking indicated domains. (F) Schematic representation of RICTOR full length (FL), and its various deletion mutants lacking indicated domains. (G) RICTOR depleted HEK293T cells were transfected with SFB vector or SFB-tagged mSIN constructs (shown in (E)) along with Myc-PTPN22. Cell lysates were pulldown with S-protein agarose beads, and their interactions were detected by immunoblotting with anti-Myc antibody. (H) mSIN depleted HEK293T cells expressing either SFB vector or SFB-tagged RICTOR constructs (shown in (F)) along with Myc-PTPN22 were lysed in 1% Triton X-100 buffer and pulldown with S-protein agarose beads. Interaction of PTPN22 with various domains was detected by immunoblotting with anti-Myc antibody. Source data are available online for this figure.

proline-rich domain of PTPN22 is mediating its interaction with RICTOR (Fig. EV4A). Also, the predicted structure displayed an interaction of N-terminus of mSIN with the N-terminal catalytic domain of PTPN22 (Fig. EV4B), and the C-terminus of RICTOR with the C-terminal domain of PTPN22 (Fig. EV4C). To biochemically confirm the interacting regions predicted through the model, we generated various PTPN22 deletion mutants (Fig. 4A). We performed mapping experiments of RICTOR or mSIN in HEK293T stable cells with knockdown of mSIN or RICTOR, respectively (Fig. EV4D,E), followed by cell lysis in Triton X-100-containing buffer to minimize the background interactions due to other mTORC2 subunits. Consistent with our predicted model, the N-terminal catalytic region of PTPN22 is essential for its binding with mSIN (Fig. 4B), while the central interdomain and the C-terminal proline-rich domain mediates its binding with RICTOR (Fig. 4C). Functionally, we tested the ability of mSIN and RICTOR binding domains of PTPN22 by individually expressing them in PTPN22 depleted cells. While the full length PTPN22 can rescue the defect in AKT phosphorylation at Ser473, the catalytic domain and the C-terminal domain fail to do so, clearly suggesting the requirement of intact PTPN22 binding with RICTOR and mSIN for mTORC2-AKT axis activation (Fig. 4D). Next, to assess the interaction interfaces on mSIN and RICTOR that mediates their interaction with PTPN22, we generated a series of mSIN and RICTOR deletion mutants that lack one or more domains (Fig. 4E,F), based on previous studies (Chen et al, 2018; Zhang et al, 2016; Zhou et al, 2015). We found that the residues 1–240 that consists of N-terminal and CRIM (conserved region in the middle) domains of mSIN are required for its association with PTPN22 (Fig. 4E,G). Previously, it was reported that the mSIN-CRIM domain exhibits a ubiquitin-like fold, which is required for the interaction and recruitment of mTORC2 substrates (Cameron et al, 2011; Tatebe et al, 2017). In order to test the impact of PTPN22 interaction with the mSIN-CRIM domain on AKT/mSIN association, we performed in vitro binding experiments using bacterially purified proteins which demonstrate that the binding of mSIN with PTPN22 did not affect the binding of AKT (mTORC2 substrate) with mSIN (Fig. EV4F,G). Additionally, competition assays in cells reveal that the interaction of AKT with mSIN remains unchanged with increasing levels of PTPN22, possibly suggesting that the binding interfaces of mSIN are different for AKT and PTPN22 (Fig. EV4H). On the other hand, a region of amino acids between 800-1001 of RICTOR, which

corresponds to armadillo like fold is required for its binding with PTPN22 (Fig. 4F,H). Together, these data indicate that PTPN22 through two distinct regions connect RICTOR and mSIN to noncatalytically assist in mTORC2 assembly and activity. Previously determined mTORC2 structures suggest that the N-terminus of mSIN lies within the vicinity of the ARM4 region of the RICTOR (Chen et al, 2018; Scaiola et al, 2020; Stuttfeld et al, 2018; Yu et al, 2022). Given the flexibility of these regions, it is tempting to speculate that PTPN22 by functioning as a linker between these two regions of mSIN and RICTOR is providing stability to this complex.

## PTPN22 promotes oncogenesis independent of its catalytic activity, in cell based and nude mice xenografts assays

Activation of mTORC2-AKT signaling axis is critical for cell survival and proliferation. We next sought to understand the biological relevance of PTPN22 on mTORC2-mediated cellular functions. We performed cell growth and colony formation assays in PTPN22 depleted cells (Fig. EV5A). We noticed that depletion of PTPN22 reduced the cell number (Fig. 5A), and colony forming ability of the cells (Fig. EV5B,C). Next, to test if PTPN22 drive these functions in a catalytic activity-independent manner, we overexpressed either wild-type or catalytic dead mutant of PTPN22 (Fig. EV5D), in PTPN22 depleted cells. Consistent with our biochemical data shown above, reintroducing wild-type as well as enzymatic dead mutant of PTPN22 rescued the defect in cell number (Fig. EV5E) and colony formation (Fig. 5B,C). Furthermore, we observed that knockdown of PTPN22 suppressed cell migration, which can be restored by exogenous expression of PTPN22 in catalytic activity-independent manner (Fig. 5D,E). On the other hand, depletion of PTPN22 using shRNA markedly enhanced apoptosis (as measured by increase in subG1 population of cells), which can be reduced back similar to control by expressing PTPN22 wild-type or catalytically dead mutant (Fig. EV5F,G). Moreover, Torin treatment prevents the rescue of apoptosis by PTPN22 overexpression in PTPN22 depleted cells (Fig. EV5H,I), possibly suggesting that the apoptotic effects of PTPN22 depletion are mediated through mTORC2 activation. Additionally, to test the effects of PTPN22 on cell proliferation, we assessed the mitotic index between PTPN22 knockdown cells in comparison to control cells by phospho-histone H3 (pH3) staining. Interestingly, the

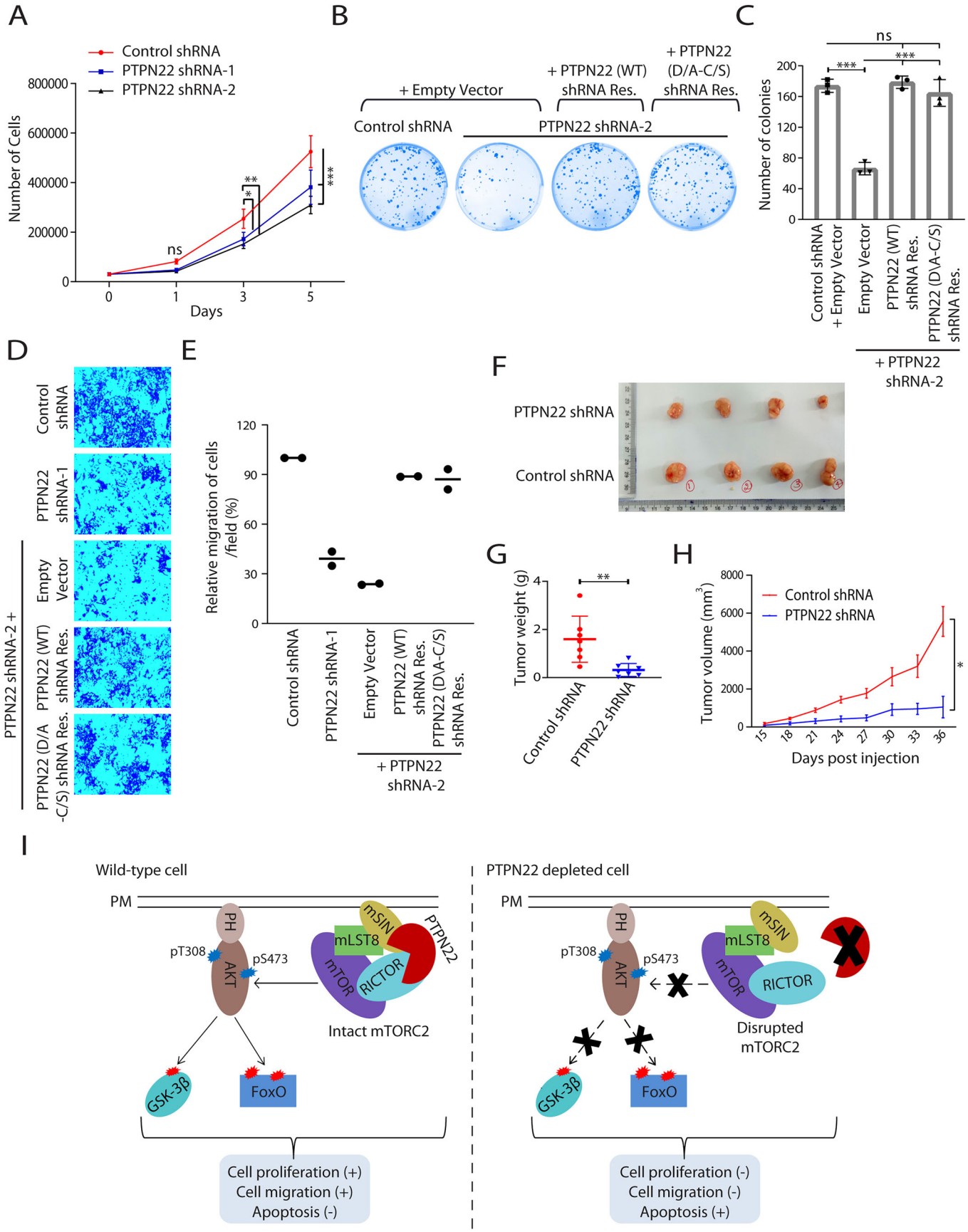

**Figure 5. PTPN22, but not its phosphatase activity, is essential for promoting oncogenesis.**

(A) Equal numbers of HCT116 cells stably expressing either control shRNA or PTPN22 shRNAs were seeded, and after day 1, cell number was measured every 2 days for the indicated durations. Quantification of cell number was calculated from three independent experiments. Error bars indicate mean ± SD ($n = 3$), *$P = 0.0151$, **$P = 0.0026$, ***$P < 0.0001$, ns: not significant (two-way ANOVA with Tukey's multiple comparisons test). (B, C) HCT116 cells expressing either control shRNA or PTPN22 shRNA were transfected with indicated constructs, and colony formation assay was performed. Images were captured after staining with crystal violet (B), and quantification data for number of colonies from three independent experiments are shown (C). Error bars indicate mean ± standard deviation ($n = 3$), ***$P = 0.00001566$ for control shRNA + empty vector v/s PTPN22 shRNA-2 + empty vector, ***$P = 0.00011134$ for PTPN22 shRNA-2 + empty vector v/s PTPN22 shRNA-2 + PTPN22 (WT) shRNA Res., ***$P = 0.00003107$ for PTPN22 shRNA-2 + empty vector v/s PTPN22 shRNA-2 + PTPN22 (D/A-C/S) shRNA Res., ns: not significant for control shRNA + empty vector v/s PTPN22 shRNA-2 + PTPN22 (WT) shRNA Res., for control shRNA + empty vector v/s PTPN22 shRNA-2 + PTPN22 (D/A-C/S) shRNA Res., and PTPN22 shRNA-2 + PTPN22 (WT) shRNA Res. v/s PTPN22 shRNA-2 + PTPN22 (D/A-C/S) shRNA Res. (one-way ANOVA with Bonferroni's multiple comparisons test). (D, E) Representative images of transwell migration assays showed the migration potential of HCT116 cells expressing indicated PTPN22 constructs, or a control vector. Phase-contrast microscopic images were taken after staining the migrated cells with crystal violet (D). Scatter plot showing the relative migration of cells/field (derived from average of four different fields) from two independent experiments are shown (E). (F–H) Knockdown of PTPN22 inhibits subcutaneous tumor growth in nude mice xenograft model. HCT116 cells stably expressing either control shRNA or PTPN22 shRNA were injected subcutaneously into athymic nude mice (Foxn1$^{-/-}$) for xenograft tumor growth. Tumor images (F), tumor weights (G), and tumor volumes (H) are shown. Error bars indicate mean ± standard deviation (for tumor weights), and mean ± standard error of mean (for tumor volumes) ($n = 7$), *$P = 0.0379$, **$P = 0.0054$ (unpaired two-tailed Student's $t$ test). (I) A proposed model to depict the role of PTPN22 in mTORC2-AKT axis activation. PTPN22 acts as a molecular bridge for mSIN-RICTOR association, thereby facilitating the proper assembly, and enhancing the kinase activity of mTORC2 towards AKT. Source data are available online for this figure.

mitotic index was significantly higher in PTPN22 knockdown cells in comparison to control shRNA cells, possibly suggesting a halt in cell cycle (Fig. EV5J). Moreover, to investigate whether PTPN22 can act as a potential oncogene in vivo, we subcutaneously transplanted HCT116 cells stably expressing either control shRNA or PTPN22 shRNA (Fig. EV5K) in nude mice. Knockdown of PTPN22 reduced the tumor formation ability in xenograft assays; as tumor volume and tumor weight were decreased as compared to control tumors (Fig. 5F–H).

In conclusion, our study identifies phosphatase PTPN22 as a non-catalytic scaffolding protein that regulates mTORC2 activity by facilitating the mSIN-RICTOR interaction (model shown in Fig. 5I). This work expands the functional repertoire of PTPN22 beyond its well-established role in immune signaling and suggests its involvement in regulating key cellular processes through mTORC2. We show that PTPN22 promotes AKT activation in both hematopoietic and non-hematopoietic cells, independent of its phosphatase activity, and contributes to oncogenesis in vitro and in vivo. In cells, mTOR forms the catalytic core of two distinct multi-subunit complexes, mTORC1 and mTORC2 (Kim and Guan, 2019; Saxton and Sabatini, 2017). Unlike the well-established regulatory mechanisms of mTORC1, only a few protein regulators have been reported to modulate mTORC2. For instance, oncogenic Ras activates mTORC2 kinase activity by directly interacting with its components at the plasma membrane (Kovalski et al, 2019). In another study, hyper-phosphorylated Rb (Retinoblastoma) protein was shown to suppress mTORC2 function by directly binding mTORC2, and inhibiting its activity (Zhang et al, 2016). However, the functional interactions between mTORC2 and phosphatases are not known. Interestingly, here we observed that a non-receptor tyrosine phosphatase, PTPN22 acts as a molecular bridge to facilitate mSIN and RICTOR binding, in enzyme independent fashion to promote mTORC2 activity and functions. Thus, our work fills an important gap of phosphatase connection with mTORC2 biology.

Previous work has suggested that PTPN22 regulates AKT activity in haematopoietic cells through indirect mechanisms, such as modulation of lipid phosphatases (Baghbani et al, 2017; Bai et al, 2023; Negro et al, 2012; Pogue et al, 2000). Our study extends these findings by providing direct evidence for a scaffolding role of PTPN22 in mTORC2, offering a new mechanistic insight into the regulation of AKT activation. While PTPN22 phosphatase activity is well studied in the context of immune regulation, its function outside of the immune system remains poorly understood. Recently, in the nematode worm *C. elegans*, PTPN22 was shown to associate with DNBP-1, a Cdc42 guanine nucleotide exchange factor, to promote epithelial cell integrity and regulate cytoskeletal machinery (Binti et al, 2024). Additionally, increased PTPN22 levels have been linked with several cancer types including esophageal carcinoma, stomach adenocarcinoma, and lymphoid and myeloid leukemias (Bottini and Peterson, 2014; Chen et al, 2020; Hebbring et al, 2013). In addition to this, various PTPN22 mutations listed in the COSMIC (Catalogue of Somatic Mutations in Cancer) database are distributed throughout the protein (Sondka et al, 2024). However, the mechanisms underlying its role in cancer are poorly understood. Given that mTOR pathway is often dysregulated in cancer, it is tempting to speculate that hyperactivation of mTORC2/AKT axis by PTPN22 might be one of the possible underlying mechanisms dysregulated in these cancer types. Collectively, our data suggest that PTPN22 may act as a potential oncogene by regulating mTORC2 assembly and functions.

## Methods

**Reagents and tools table**

| Reagent/resource | Reference or source | Identifier or catalog number |
|---|---|---|
| **Recombinant DNA** | | |
| pRK5-myc-mTOR | Addgene | Cat#1861 |
| pDONOR201-RICTOR | This study | N/A |
| pDONOR201-RAPTOR | This study | N/A |
| pDONOR201-mSIN1 | This study | N/A |
| pDONOR201-mLST8 | This study | N/A |
| pDONOR201-PRAS40 | This study | N/A |
| pDONOR201-PTPN22 | This study | N/A |

| Reagent/resource | Reference or source | Identifier or catalog number |
|---|---|---|
| pDONOR201-PTPN5 | This study | N/A |
| pDONOR201-AKT | This study | N/A |
| psPAX2 | Addgene | Cat#12260 |
| pMD2.G | Addgene | Cat#12259 |
| **Antibodies** | | |
| Rabbit anti-PTPN22 | Cell Signaling Technologies | Cat#14693 |
| Rabbit anti-mTOR | Cell Signaling Technologies | Cat#2972 |
| Rabbit anti-mTOR | Cell Signaling Technologies | Cat#2983 |
| Rabbit anti-RICTOR | Cell Signaling Technologies | Cat#9476 |
| Rabbit anti-RAPTOR | Cell Signaling Technologies | Cat#2280 |
| Rabbit anti-mLST8/GβL | Cell Signaling Technologies | Cat#3274 |
| Rabbit anti-SIN1 | Abcam | Cat#ab71152 |
| Rabbit anti-AKT | Cell Signaling Technologies | Cat#9272 |
| Rabbit anti-pS473-AKT | Cell Signaling Technologies | Cat#9271 |
| Rabbit anti-pS473-AKT | Cell Signaling Technologies | Cat#9018 |
| Rabbit anti-pT308-AKT | Cell Signaling Technologies | Cat#9275 |
| Rabbit anti-GSK-3β | Cell Signaling Technologies | Cat#12456 |
| Rabbit anti-pS9-GSK-3β | Cell Signaling Technologies | Cat#5558 |
| Rabbit anti-FoxO1 | Cell Signaling Technologies | Cat#2880 |
| Rabbit anti-pT24-FoxO1/pT32-FoxO3a | Cell Signaling Technologies | Cat#9464 |
| Rabbit anti-p70S6K | Cell Signaling Technologies | Cat#2708 |
| Rabbit anti-pT389-p70S6K | Cell Signaling Technologies | Cat#9205 |
| Rabbit anti-PKCα | Cell Signaling Technologies | Cat#2056 |
| Rabbit anti-phospho-PKCα/βII (pT638/641) | Cell Signaling Technologies | Cat#9375 |
| Rabbit anti-SGK1 | Abcam | Cat#ab59337 |
| Rabbit anti-pS422-SGK1 | Abcam | Cat#ab55281 |
| Rabbit anti-pS10-Histone H3 | Cell Signaling Technologies | Cat#9701 |
| Mouse anti-β-Actin | Sigma-Aldrich | Cat#A5441 |
| Mouse anti-Flag | Sigma-Aldrich | Cat#F3165 |
| Mouse anti-c-Myc (9E10) | Santa Cruz Biotechnology | Cat#sc40 |
| Mouse anti-c-Myc (9E10) | Sigma-Aldrich | Cat#M4439 |

| Reagent/resource | Reference or source | Identifier or catalog number |
|---|---|---|
| Mouse anti-MBP | Sigma-Aldrich | Cat#M1321 |
| Rabbit anti-6x-His | Bethyl Laboratories | Cat#A190-114A |
| Mouse anti-GST | Santa Cruz Biotechnology | Cat#sc-138 |
| HRP-conjugated Rabbit Anti-Mouse | Jackson Immunologicals | Cat#315-035-048 |
| HRP-conjugated Goat Anti-Rabbit | Jackson Immunologicals | Cat#111-035-144 |
| HRP-conjugated Anti-Mouse TrueBlot | Rockland Immunochemicals | Cat#18-8817-33 |
| HRP-conjugated Anti-Rabbit TrueBlot | Rockland Immunochemicals | Cat#18-8816-33 |
| Goat Anti-Rabbit IgG (H + L) Alexa Fluor™ 488 | Invitrogen | Cat#A-11034 |
| **Experimental models: cell lines** | | |
| HEK293T | ATCC | N/A |
| BOSC23 | ATCC | N/A |
| Jurkat | ATCC | N/A |
| HCT116 | ATCC | N/A |
| U-2 OS | ATCC | N/A |
| HeLa | ATCC | N/A |
| **Experimental models: organisms/strains** | | |
| Athymic nude female mice (Foxn1$^{-/-}$) (5–6 weeks old) | This study | Institutional experimental animal facility |
| **Oligonucleotides and other sequence-based reagents** | | |
| PTPN22 shRNAs targeting sequences 5′-CTAGTGCTCTTGGTGT ATATT-3′ 5′-GAATTGATACAGCAG AGAGAA-3′ | Open Biosystems | TRCN0000003037 TRCN0000003038 |
| mSIN shRNAs targeting sequences 5′-GTTGGGACTTTGGT ATTAGAA-3′ 5′-GCCACAGTACAGGA TATGCTT-3′ | Dharmacon | TRCN0000003152 TRCN0000003153 |
| RICTOR shRNAs targeting sequences 5′-CGTCGGAGTAACCAA AGATTA-3′ 5′-GCACCCTCTATTGCTA CAATT-3′ 5′-CGGAGGTTCATACAAG AATTA-3′ | Dharmacon | TRCN0000074291 TRCN0000074290 TRCN0000074289 |
| PTPN22 sgRNAs targeting sequences 5′-GGAGTTTGCCAATGAA TTTC-3′ 5′-GGATGTACGTTGTTACC AAG-3′ 5′-CAAAACCTATCCTACAA CTG-3′ | This study | N/A |

| Reagent/resource | Reference or source | Identifier or catalog number |
|---|---|---|
| Forward RT-PCR PTPN22_primer 5'-AGCTAGTTTTGCACCCTGCTA-3' | This study | N/A |
| Reverse RT-PCR PTPN22_primer 5'-TTGGTCCGTGTTATTGGCACC-3' | This study | N/A |
| Forward RT-PCR GAPDH_primer 5'-CGACCACTTTGTCAAGCTCA-3' | This study | N/A |
| Reverse RT-PCR GAPDH_primer 5'-AGGGGAGATTCAGTGTGGTG-3' | This study | N/A |
| **Chemicals, enzymes and other reagents** | | |
| Phorbol 12-myristate 13-acetate (PMA) | MedChemExpress | Cat#HY-18739 |
| PTPN22-IN-1 | MedChemExpress | Cat#HY-139693 |
| Torin1 | Tocris | Cat#4247 |
| Polyethylenimine (PEI) | Polysciences | Cat#23966 |
| TurboFect transfection reagent | Invitrogen | Cat#R0531 |
| Lipofectamine LTX reagent with Plus reagent | Invitrogen | Cat#15338030 |
| RPMI 1640 medium | HiMedia | Cat#AL028A |
| Fetal bovine serum (FBS) | Gibco | Cat#10270-106 |
| DMEM | Gibco | Cat#10313-021 |
| L-glutamine | Sigma-Aldrich | Cat#25030149 |
| Phosphate-buffered saline (PBS) | HyClone | Cat#SH30256.01 |
| Aprotinin | Sigma-Aldrich | Cat#A1153 |
| Pepstatin A | GoldBio | Cat#P-020-5 |
| Bovine serum albumin (BSA) | Sigma-Aldrich | Cat#A2153 |
| Epidermal Growth Factor (EGF) | Invitrogen | Cat#53003-018 |
| Penicillin/streptomycin | GeneDireX | Cat#CC502-0100 |
| Polybrene | Sigma-Aldrich | Cat#H9268 |
| Puromycin | Gibco | Cat#A11138-02 |
| Pierce™ Bradford plus protein assay kit | ThermoFisher Scientific | Cat#23236 |
| Protein G sepharose beads | Cytiva | Cat#17061801 |
| Streptavidin sepharose beads | Cytiva | Cat#17511301 |
| S-protein agarose beads | Millipore | Cat#69704 |
| PVDF membranes | Millipore | Cat#IPVH00010 |
| Microsep centrifugal filters (MWCO#10 K) | Pall Life Sciences | Cat#MCP010C41 |
| Gel filtration HMW calibration kit | Cytiva | Cat#28-4038-42 |
| RNA isolation kit | Macherey-Nagel | Cat#740955 |
| PrimeScript™ cDNA synthesis kit | Takara Bio | Cat#6110B |

| Reagent/resource | Reference or source | Identifier or catalog number |
|---|---|---|
| Glutathione sepharose™ 4B beads | GE Healthcare | Cat#GE17-0756-01 |
| Dextran sepharose™ beads | GE Healthcare | Cat#GE28-9355-97 |
| Nickel NTA agarose beads | Qiagen | Cat#30210 |
| Biotin | Sigma-Aldrich | Cat#B4501 |
| Paraformaldehyde | Sigma-Aldrich | Cat#158127 |
| Crystal violet | Sigma-Aldrich | Cat#C3886 |
| RNase A | Sigma-Aldrich | Cat#R4875 |
| Propidium Iodide | Sigma-Aldrich | Cat#P4864 |
| Polyethylene terephthalate membrane filters | Falcon, BD Biosciences | Cat#353097 |
| Matrigel | Corning | Cat#356234 |
| 0.45 μm Filters | Millipore | Cat#SLHV033RS |
| **Software** | | |
| Fiji/ImageJ 1.54f software | NIH, Bethesda, MD, USA | https://imagej.nih.gov/ij |
| Cytoscape 3.10.2 | Cytoscape | https://cytoscape.org/ |
| GraphPad Prism 8 | GraphPad | Graphpad.com |
| Adobe Photoshop CS3 | Adobe | Adobe Creative Studio |
| Adobe Illustrator CS3 | Adobe | Adobe Creative Studio |
| Zen 3.4 Blue | Zeiss | https://www.zeiss.com/ |
| CHOP-CHOP web tool | Labun et al (2016) | https://chopchop.cbu.uib.no |
| Alpha Fold Server | Abramson et al (2024) | https://alphafoldserver.com |
| Flowjo_V10 | BD | https://www.flowjo.com/ |
| **Other** | | |
| Gel Doc™ XR system with Image Lab™ software | Bio-Rad | N/A |
| AKTA-FPLC system | GE HealthCare | N/A |
| Zeiss Elyra 9 | Zeiss | N/A |
| Chemidoc system | UV Tech, Cambridge | N/A |
| Inverted microscopy, IX73 | Olympus | N/A |
| BD Accuri™ C6 | BD Biosciences | N/A |
| BD LSRFortessa™ X-20 Cell Analyzer | BD Biosciences | N/A |

## Antibodies and reagents

Immunoprecipitation (IP) and western blotting (WB) were performed by using the following antibodies: PTPN22 (CST 14693S; WB 1:1000 or IP 1:100 or 1:1000), mTOR (CST 2972S; WB 1:1000), mTOR (CST 2983S; WB 1:1000), RICTOR (CST 9476S; WB 1:1000), RAPTOR (CST 2280S; WB 1:1000), mLST8/

GβL (CST 3274S; WB 1:1000), SIN1 (abcam ab71152; WB 1:3000 or IP 2.5 μg), AKT (CST 9272S; WB 1:2000), pS473-AKT (CST 9271L; WB 1:1500), pS473-AKT (CST 9018S; WB 1:1500), pT308-AKT (CST 9275L; WB 1:1000), GSK-3β (CST 12456T; WB 1:2000), pS9-GSK-3β (CST 5558T; WB 1:1000), FoxO1 (CST 2880T; WB 1:1000), pT24-FoxO1/pT32-FoxO3a (CST 9464T; WB 1:1000), p70S6K (CST 2708S; WB 1:2000), pT389-p70S6K (CST 9205S; WB 1:1000), PKCα (CST 2056T; WB 1:1000), phospho-PKCα/βII [pT638/641] (CST 9375T; WB 1:1000), SGK1 (ab59337; WB 1:1000), pS422-SGK1 (ab55281; WB 1:500), pS10-Histone H3 (CST 9701L; IF 1:200), β-actin (Sigma-Aldrich A5441; WB 1:10,000), FLAG (Sigma-Aldrich F3165; WB 1:10,000 or IP 2 μg), c-Myc (Santa Cruz Biotechnologies clone 9E10 (sc-40); WB 1:1000), c-Myc (Sigma-Aldrich M4439; IP 1 μg), MBP (Sigma-Aldrich M1321; WB 1:5000), anti-6-His (Bethyl A190-114A; 1:800) and GST (Santa Cruz Biotechnologies sc-138; WB 1:2500). Secondary antibodies: HRP-conjugated anti-mouse (315-035-048) and anti-rabbit (111-035-144) antibodies were obtained from Jackson Immunologicals. TrueBlot secondary antibodies: HRP-conjugated anti-mouse TrueBlot (18-8817-33) and anti-rabbit TrueBlot (18-8816-33) antibodies were purchased from Rockland Immunochemicals. Alexa Fluor™ 488 (A-11034) was purchased from Invitrogen. Phorbol 12-myristate 13-acetate (PMA) (HY-18739) and PTPN22 inhibitor, PTPN22-IN-1 (HY-139693) were purchased from MedChemExpress. Torin1 (4247) was purchased from Tocris.

## Cell culture and transfections

HEK293T, Jurkat, HCT116, HeLa, U-2 OS and BOSC23 cell lines were used in this study. HEK293T, U-2 OS, HeLa and Jurkat cell lines were maintained in RPMI 1640 medium (HiMedia, AL028A) supplemented with 10% fetal bovine serum (FBS; Gibco, 10270-106) and 1% penicillin/streptomycin (GeneDireX, CC502-0100). HCT116 and BOSC23 cell lines were grown in DMEM (Gibco, 10313-021) supplemented with 10% fetal bovine serum, 2 mM L-glutamine (Sigma-Aldrich, 25030149) and 1% penicillin and streptomycin. All cell lines were continuously monitored by microscopy for their original morphology and also regularly tested for mycoplasma contamination by using 4', 6-diamidino-2-phenylindole (DAPI) staining. For transfection of various plasmids in HEK293T or BOSC23 cells polyethylenimine (PEI; Polysciences, 23966) was used according to the manufacturer's protocol. Plasmids were transfected into HCT116 and HeLa cells using TurboFect transfection reagent (Invitrogen, R0531) according to the manufacturer's protocol. Briefly, the plasmids were diluted in serum-free medium was mixed with PEI (1 μg/μl)/TurboFect at a ratio of 1:3 (DNA amount: Transfection reagent volume), followed by incubation for 15–20 min at room temperature and then added to cells. Jurkat cells were transfected using Lipofectamine LTX reagent with Plus reagent (Invitrogen, 15338030) according to the manufacturer's protocol. At 6-8 h post-transfection, the culture medium was replaced with fresh medium, and cells were harvested 24–36 h after transfection for further analyses.

For serum starvation and re-stimulation with growth factors to assess mTORC2 activation, subconfluent cells were incubated with serum-free media for 16 h, followed by stimulation with 100 ng/ml EGF (Invitrogen, 53003-018) for indicated time points.

## Plasmids and shRNAs

pRK5-myc-mTOR (Addgene, #1861) plasmid was purchased from Addgene. All cDNAs (RICTOR, RAPTOR, mSIN1, mLST8, PRAS40, PTPN22, AKT, PTPN5) were amplified by PCR and cloned into donor vector pDONOR201 (Invitrogen) using gateway cloning, and then moved to triple-tagged (S-protein/Flag/streptavidin-binding protein) SFB, Myc, glutathione S-transferase (GST), 6X-His, and maltose-binding protein (MBP)-tagged destination vectors (Invitrogen) for expression. All clones were verified by Sanger sequencing. Point mutants, PTPN22, mSIN1 and RICTOR domain deletions and the shRNA-resistant mutants were generated by PCR-based site-directed mutagenesis in pDONOR201, and then cloned into SFB, GST and Myc-tagged destination vectors. Lentiviral-based shRNAs for PTPN22 (shRNA#1: 5'-CTAGTGCTCTTGGTGTATATT-3' [TRCN0000003037], shRNA#2: 5'-GAATTGATACAGCAGAGA-GAA-3' [TRCN0000003038]) were purchased from Open Biosystems. Lentiviral-based shRNA plasmids for mSIN (shRNA#1: 5'-GTTGGGACTTTGGTATTAGAA-3' [TRCN0000003152], shRNA#2: 5'-GCCACAGTACAGGATATGCTT-3' [TRCN0000003153]), and RICTOR (shRNA#1: 5'-CGTCGGAGTAACCAAAGAT TA-3' [TRCN0000074291], shRNA#2: 5'-GCACCCTCTATTGCTA-CAATT-3' [TRCN0000074290], shRNA#3: 5'-CGGAGGTTCATA-CAAGAATTA-3' [TRCN0000074289]) were purchased from Dharmacon. CRISPR sgRNA oligos for PTPN22 were designed using CHOP-CHOP web tool. The sequences for the guide RNAs were as follows: (sgRNA#1: 5'-GGAGTTTGCCAATGAATTTC-3', sgRNA#2: 5'-GGATGTACGTTGTTACCAAG-3', sgRNA#3: 5'-CAAAACC-TATCCTACAACTG-3'). Double-stranded DNA oligos that encode single sgRNAs were annealed and then cloned into BsmBI restriction sites of the LentiCRISPRv2 vector. All clones were verified by Sanger sequencing.

## Lentiviral production and RNA interference

Lentivirus-based pLKO.1 vectors encoding PTPN22, mSIN, RIC-TOR or a scrambled shRNA were co-transfected along with the psPAX2 packaging and pMD2.G envelope plasmids (Addgene, 12260 and 12259) in BOSC23 or HEK293T packaging cells using PEI. Viral supernatants were collected 48 and 72 h post-transfection, filtered through 0.45 μm filters (Millipore, SLHV033RS). Target cells were infected twice with viral medium along with polybrene (8 μg/ml; Sigma-Aldrich, H9268) for 36 h. After transduction, cells were collected and processed for various assays, and immunoblotting was performed with specific antibodies to check the efficiency of knockdown. Stable cell lines were generated by replacing culture medium with fresh medium containing puromycin (2 μg/ml; Gibco, A11138-02), 36 h after the second transduction, followed by clonal selection (5–10 days). The positive clones were confirmed by immunoblotting with specific antibodies. Lentiviral-based transduction in Jurkat cells was performed as described in (Kory et al, 2020). Briefly, cells were cultured in RPMI 1640 medium containing polybrene (8 μg/ml), and then transduced with lentiviral particles by centrifugation at 2200 rpm for 1 h at 37 °C. After incubation for 18 h, viral media was removed and cell pellets were washed twice in phosphate-buffered saline (PBS) (HyClone, SH30256.01), then re-seeded into fresh RPMI 1640 medium having puromycin (1 μg/ml). After 3–7 days post-

infection, cells were collected and processed for various assays, and immunoblotting. For mSIN and RICTOR knockdown experiments, multiple shRNAs viral medium was pooled to obtain better knockdown efficiency.

## Generation of knockout cell lines

LentiCRISPRv2 vector encoding PTPN22 guide RNAs or a control sgRNA were co-transfected along with the psPAX2 packaging and pMD2.G envelope plasmids (Addgene, 12260 and 12259) in BOSC23 packaging cells using PEI. Viral supernatants were collected 48 h post-transfection, filtered through 0.45 μm filters (Millipore, SLHV033RS). Target cells were infected twice with viral medium along with polybrene (8 μg/ml; Sigma-Aldrich, H9268) for 36 h. Stable cell lines were generated by replacing culture medium with fresh medium containing puromycin (2 μg/ml; Gibco, A11138-02), 36 h after the second transduction, followed by clonal selection. The positive clones were confirmed by immunoblotting for the absence of protein expression with specific antibodies.

## Cell lysis, immunoprecipitation, pulldown assays, and western blotting

To analyze various protein levels in the whole cell lysates, cells were lysed with NETN buffer as described above. The soluble fractions were isolated by centrifugation at $15,871 \times g$ for 15 min at 4 °C. Total protein concentration of the samples was measured using Pierce™ Bradford plus protein assay kit (Thermo Fisher Scientific, 23236), and were boiled after addition of 6X SDS sample buffer at 100 °C for 5 min. For immunoprecipitation and pulldown assays, cells were rinsed once in ice-cold PBS and lysed either in 0.3% CHAPS lysis buffer [40 mM HEPES pH 7.4, 120 mM NaCl, 1 mM EDTA and 0.3% (w/v) CHAPS (3-((3-cholamidopropyl) dimethylammonio)-1-propanesulfonate)] or in 1% Triton X-100 lysis buffer (20 mM Tris/HCl pH 8.0, 100 mM NaCl, 1 mM EDTA and 1% (v/v) Triton X-100) supplemented with 1 μg/ml pepstatin A, 1 μg/ml aprotinin and 100 mM PMSF (phenylmethylsulfonyl fluoride) on ice for 20 min. Cell debris was removed by centrifugation at $15,871 \times g$ for 15 min at 4 °C, and protein concentrations were determined as mentioned above. Equal amount of total protein from each lysate was incubated with 1–2.5 μg of specified antibody bound to either protein A/ G sepharose beads (Cytiva, 17096303/17061801), or streptavidin sepharose beads (Cytiva, 17511301), or S-protein agarose beads (Millipore, 69704) for 3–6 h at 4 °C. The bound protein complexes were washed five times with lysis buffer and eluted by boiling in 2 × SDS sample buffer at 100 °C for 5 min. The eluates were then resolved by SDS-PAGE, transferred on PVDF membranes (Millipore, IPVH00010) and immunoblotting was performed following standard protocols. Protein bands were visualized using enhanced chemiluminescence (ECL) based detection (Amersham, RPN2236). Densitometry based quantitative analyses were done by Fiji ImageJ software (NIH, Bethesda, MD, USA) when needed.

## Size-exclusion chromatography

Gel-filtration experiments were adapted as described previously (Liu et al, 2013). In brief, Jurkat cells were cultured in 10 cm dishes (four dishes per gel filtration experiment). Cells were pellet down, and washed with phosphate-buffered saline (PBS), lysed in 0.3%

CHAPS buffer (as mentioned above) on ice for 30 min, and insoluble fractions were removed by centrifugation at $15,871 \times g$ for 15 min at 4 °C. Supernatant was filtered through a 0.22 μm filter and concentrated to 1 ml ( ~ 8 mg) using microsep centrifugal filters (MWCO#10 K; MCP010C41, Pall Life Sciences). Before run, the Superdex 200 column (GE HealthCare) was calibrated using gel filtration HMW calibration kit (Cytiva, 28-4038-42) that contains: Blue dextran 2000 (to determine void volume) ovalbumin (44 kDa), conalbumin (75 kDa), aldolase (158 kDa), ferritin (440 kDa) and thyroglobulin (669 kDa). Lysates were injected onto the column and chromatography was performed on the AKTA-FPLC system (GE HealthCare) with CHAPS buffer. In total, 500 μl of fractions were collected and samples were concentrated to a final volume of 100 μl, and indicated fractions were resolved by SDS-PAGE followed by immunoblotting with specific antibodies.

## RNA isolation and reverse transcription polymerase chain reaction (RT-PCR)

Total RNA was isolated from control and PTPN22 knockdown cell lines using Nucleospin RNA isolation kit (Macherey-Nagel, 740955) following the manufacturer's instructions. 1 μg of total RNA was reverse transcribed using PrimeScript™ cDNA synthesis kit (Takara Bio, 6110B). The synthesized cDNA products were amplified through PCR using respective primers.

PTPN22 forward: 5'-AGCTAGTTTTGCACCCTGCTA-3'
PTPN22 reverse: 5'-TTGGTCCGTGTTATTGGCACC-3'
GAPDH forward: 5'-CGACCACTTTGTCAAGCTCA-3'
GAPDH reverse: 5'-AGGGGAGATTCAGTGTGGTG-3'.

The PCR products were separated by 2% agarose gel electrophoresis and stained with ethidium bromide (Sigma-Aldrich, E7637). The gels were visualized by Gel Doc™ XR system with Image Lab™ software (Bio-Rad). GAPDH was used as a control to normalize the PTPN22 expression levels.

## Recombinant protein purification and in vitro binding assays

GST, GST-mSIN, GST-AKT (K179D), GST-PTPN22, His-tagged PTPN22, MBP or MBP-mSIN, MBP-PTPN22 full-length proteins were expressed in *Escherichia coli* BL21 (DE3) cells, and the proteins were affinity purified using either glutathione sepharose™ 4B beads (GE Healthcare, GE17-0756-01) [for GST-tagged proteins] or dextran sepharose™ beads (GE Healthcare, GE28-9355-97) [for MBP-tagged proteins] or Nickel NTA agarose beads (Qiagen, 30210) for 2 h at 4 °C. Then, beads were washed five times with wash buffer (50 mM Tris pH 7.5, 300 mM NaCl, 0.01% IGEPAL, 1 mM DTT, and protease inhibitors) and bound proteins were eluted with the elution buffer containing 50 mM Tris (pH 8.0), 150 mM NaCl, and 10 mM reduced glutathione (GST-tagged proteins), 20 mM Tris (pH 7.5), 200 mM NaCl, 1 mM EDTA, and 10 mM maltose (MBP-tagged proteins) or 50 mM Tris (pH 7.5), 150 mM NaCl, 300 mM imidazole with 0.1% IGEPAL (His-tagged proteins) for 60 min at 4 °C.

For purification of RICTOR, either GST-RICTOR full-length protein were expressed in *Escherichia coli* BL21 (DE3) cells, after sonication the cell lysates were cleared by centrifugation at $15,871 \times$

g for 45 min at 4 °C. Cell lysates were then concentrated using centrifugal filters (MWCO#10 K; Millipore) or HEK293T cells having stable knockdown of mSIN were transfected with SFB-RICTOR. On the day after transfection, cells were collected and lysed in 1% Triton X-100 lysis buffer as described above. Cell lysates were clarified by centrifugation at $15,871 \times g$ for 20 min at 4 °C, and incubated with streptavidin sepharose beads (Cytiva, 17511301) for 1 h at 4 °C followed by washing with high salt buffer [NETN lysis buffer having 200 mM NaCl] (10 times), and then with low salt buffer [NETN lysis buffer having 50 mM NaCl] (five times). The purified SFB-RICTOR was eluted with 2 mg/ml biotin (Sigma-Aldrich, B4501) for 90 min at 4 °C.

For in vitro binding of mSIN and PTPN22, equal amounts of the purified GST or GST-mSIN proteins immobilized to glutathione sepharose™ 4B beads were incubated with bacterially purified MBP-PTPN22 (2–2.5 μg) for 2 h at 4 °C. For RICTOR and PTPN22 binding, equal amounts of the purified MBP or MBP-PTPN22 proteins immobilized to dextran sepharose™ 4B beads were incubated with concentrated bacterial cell lysates expressing GST-RICTOR for 1 h at 4 °C. The beads were washed five times with wash buffer and bound proteins were eluted by boiling in 2× SDS sample buffer at 100 °C for 5 min. The proteins were resolved by SDS-PAGE, and the interactions were analyzed by western blotting. For in vitro binding experiments of mSIN with AKT, equal amounts of the purified MBP or MBP-mSIN proteins immobilized to dextran sepharose™ 4B beads were incubated with bacterially purified GST-AKT (0.5 μg) either in the presence of recombinant His-PTPN22 or equal volume of corresponding buffer for 1 h at 4 °C. The beads were washed five times with wash buffer and bound proteins were eluted by boiling in 2× SDS sample buffer at 100 °C for 5 min. The proteins were resolved by SDS-PAGE, and the interactions were analyzed by western blotting. For in vitro binding of mSIN with RICTOR, equal amounts of the purified GST or GST-mSIN proteins immobilized to glutathione sepharose™ 4B beads were incubated with purified SFB-RICTOR either in the presence of recombinant His-PTPN22 or equal volume of corresponding buffer for 1 h at 4 °C. The beads were washed five times with wash buffer and bound proteins were eluted by boiling in 2× SDS sample buffer at 100 °C for 5 min. The proteins were resolved by SDS-PAGE, and the interactions were analyzed by western blotting with anti-Flag antibody.

## In vitro kinase assays

For the mTORC2 in vitro kinase assays, HCT116 cells stably expressing control shRNA and PTPN22 shRNA were transfected with SFB-RICTOR. Forty-eight hours post-transfection, RICTOR was affinity purified with streptavidin sepharose beads (Cytiva, 17511301) in 0.3% CHAPS lysis buffer, followed by elution with 2 mg/ml biotin (Sigma-Aldrich, B4501), and used as a kinase source for the assays. Bacterially purified 1 μg of GST-AKT (K179D) was incubated with SFB-RICTOR immunoprecipitates and 200 μM ATP in the kinase assay buffer (25 mM HEPES pH 7.4, 100 mM potassium acetate, 1 mM $MgCl_2$) at 30 °C for 1 h. The reactions were stopped by the addition of 2× SDS sample buffer, and the samples were resolved by SDS-PAGE. Levels of AKT phosphorylation were detected by immunoblotting using the pS473-AKT antibody.

## Colony formation assays

Cells were seeded in six-well plates at a density of 1000 cells/well [for knockdown experiments] or 500 cells/well [for rescue experiments] in 2 ml complete DMEM medium (in duplicates) and cultured for 10–14 days until the colonies were formed. Cells were washed twice with phosphate-buffered saline (PBS) (HyClone, SH30256.01), and then fixed with 4% paraformaldehyde (PFA; Sigma-Aldrich, 158127) for 15 min at room temperature. After washing with distilled water, cells were stained with 0.5% (w/v) crystal violet (Sigma-Aldrich, C3886) for 15 min and then again washed with water twice to remove unbound dye. The cells were air-dried and the number of colonies were counted and imaged under Gel Doc™ XR system with Image Lab™ software (Bio-Rad). All assays were performed independently at least three times.

## Cell proliferation assays

Cells were transfected with indicated constructs. Seventy-two hours post-transfection, equal numbers of cells were seeded in four different plates (of each type) in complete DMEM medium. After day 1, cells were trypsinized and counted regularly for the indicated durations using Neubauer chamber (Hausser Scientific) according to the manufacturer's instructions. Quantification of cell number was calculated from three independent experiments.

## Measuring sub-G1 population

Cells were harvested by centrifugation at $200 \times g$ for 5 min at 4 °C and the cell pellets were resuspended in 300 μl of phosphate-buffered saline (PBS) followed by addition of 700 μl of ice-cold ethanol (100%) dropwise. The cells were fixed on ice for 30 min. DNA fragments were extracted with 0.2 M phosphate-citrate buffer (pH 7.8) and 100 μg/ml RNase A (Sigma-Aldrich, R4875) at 37 °C for 30 min, then stained with 200 μl Propidium Iodide (50 μg/ml; Sigma-Aldrich, P4864) at 37 °C for 30 min. Cells were gated to exclude the debris and the sub-G1 population was analyzed by flow cytometry (BD Accuri™ C6, BD Biosciences or BD LSRFortessa™ X-20 Cell Analyzer, CA, USA). Assays were performed independently either two or three times.

## Cell migration assays

Transwell migration assays were performed as described previously (Kumar et al, 2019). Briefly, cells from different experimental conditions were seeded in the upper chamber of transwell inserts (24-well, 8 μm polyethylene terephthalate membrane filters; Falcon, BD Biosciences, 353097) at $1.0 \times 10^5$ cells per well, in 200 μl of serum-free DMEM. The bottom chamber contained DMEM with 10% FBS as a chemoattractant. Cells were allowed to migrate for 24 h. After the incubation period, non-migrant cells on the upper side of the filter were removed by a cotton swab. Filters were then washed with phosphate-buffered saline (PBS) and fixed with 4% paraformaldehyde for 15 min at room temperature, and inserts were stained with 0.1% crystal violet for 20 min. The migrated cells were imaged using an inverted microscope (Olympus, IX73). Quantification of migrated cells per field was calculated from two independent experiments (derived from average of four different fields).

## Immunofluorescence

Cells from different experimental conditions were grown on glass coverslips and fixed with 2% paraformaldehyde solution in PBS for 15 min at room temperature. Then cells were washed with 1× PBS three times, and permeabilized with 0.2% Triton X-100 buffer at room temperature for 5 min. After this step, cells were washed thrice with 1X PBS, and were incubated with 5% BSA for blocking at room temperature for 60 min. After incubation, cells were incubated with primary antibody overnight at 4 °C followed by washing thrice with 1× PBS. Cells were then incubated with Alexa Fluor™ 488 secondary antibody at 37 °C for 60 min followed by washing thrice with 1× PBS. Nuclei were stained with 4′,6-diamidino-2-phenylindole (DAPI), followed by washing with 1× PBS three times, and coverslips were mounted with glycerol containing paraphenylenediamine and imaged using Carl Zeiss ELYRA Super-Resolution Microscope with Plan apochromat 63×/1.4 DIC M27 oil immersion lens. Raw images were acquired with Zen Black software using Z-stacks with a step size of 0.5 μm and processed through Structured Illumination Microscopy (SIM) module using Zen Blue software.

## Nude mice xenograft assays

All animal xenograft experiments were conducted following approval from CDFD institutional animal ethics committee (approved by CPCSEA, GOI), and performed at the institutional experimental animal facility (EAF). Briefly, HCT116 cells $(5.0 \times 10^6)$ of each subline were trypsinized and washed with phosphate-buffered saline (PBS). The cell suspensions were mixed with Matrigel (1:1 ratio, v/v) (Corning, 356234) and injected subcutaneously into athymic nude female mice (Foxn1$^{-/-}$) (5–6 weeks old). Tumor size was measured every 3 days with a digital caliper, and the tumor volumes were calculated with formula V = 0.5 x Length (L) x Width$^2$ (W), where L is the longest diameter and W is the shortest diameter. On the last day (45 day), the mice were euthanized, and the xenograft tumors were dissected and the weights were noted.

## Statistical analysis

Statistical analyses were performed by two-tailed unpaired Student's *t* tests for two samples and one-way analysis of variance (ANOVA) for multiple samples and further corrected by Bonferroni's multiple comparisons test using GraphPad Prism 8.0 (GraphPad Software Inc., San Diego, CA, USA). For cell proliferation assays (Figs. 5A and EV5E), two-way analysis of variance (ANOVA) corrected with Tukey's multiple comparisons test was performed. Error bars either represented as the mean ± standard error of mean (SEM) or mean ± standard deviation (SD) as mentioned in the figure legends. The significance levels are indicated by asterisks: $*P < 0.05$, $**P < 0.01$, $***P < 0.001$, $****P < 0.0001$ and *ns*: not significant. The exact *P* values are provided in the respective figure legends.

## Data availability

This study includes no data deposited in external repositories.

The source data of this paper are collected in the following database record: biostudies:S-SCDT-10_1038-S44319-025-00576-5.

## Peer review information

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

## Acknowledgements

This work was supported by DBT/Wellcome Trust India Alliance Senior Fellowship grant (IA/S/16/2/502729 to SM) and CDFD core funds. KG acknowledges the fellowship support from the Council of Scientific and Industrial Research (CSIR) (09/724(0142)/2020 EMR-I), India. The authors thank Nanci Rani for providing technical assistance, sophisticated equipment facility and experimental animal facility at CDFD for assistance in sequencing and animal experiments, respectively. We thank all members of the LCDCS, CDFD for their critical inputs.

## Author contributions

**Keshav Gupta**: Conceptualization; Data curation; Formal analysis; Investigation; Methodology; Writing—original draft; Writing—review and editing. **Nagalakshmi Kommineni**: Formal analysis; Investigation; Methodology. **Tanuja Bogadi**: Formal analysis; Investigation; Methodology. **Neeraja P Alamuru-Yellapragada**: Formal analysis; Investigation; Methodology. **Subbareddy Maddika**: Conceptualization; Data curation; Formal analysis; Supervision; Funding acquisition; Writing—original draft; Project administration; Writing—review and editing.

Source data underlying figure panels in this paper may have individual authorship assigned. Where available, figure panel/source data authorship is listed in the following database record: biostudies:S-SCDT-10_1038-S44319-025-00576-5.

## Disclosure and competing interests statement

The authors declare no competing interests.

# Expanded View Figures

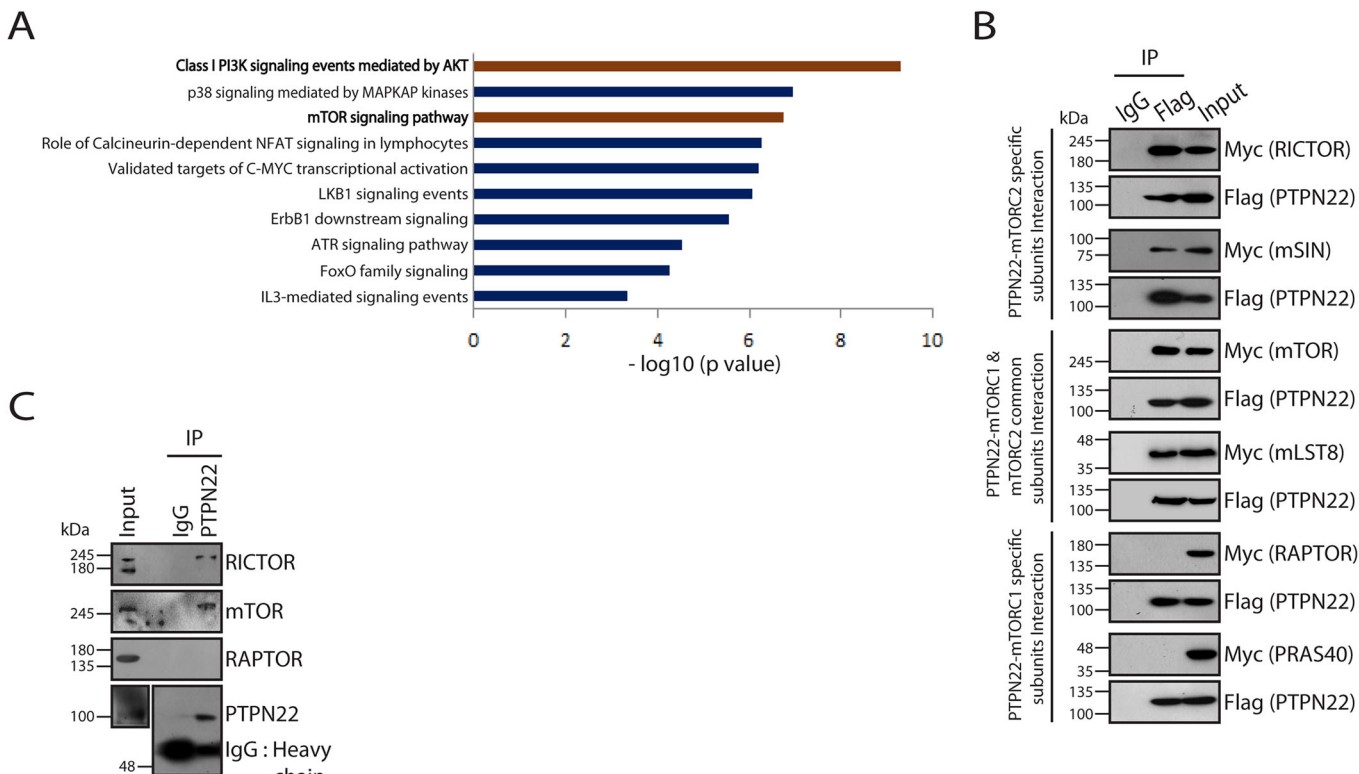

**Figure EV1. PTPN22 specifically binds to mTORC2 components, related to Fig. 1.**

(A) Bar graphs showing the −log10 *P* values of the most enriched pathways in the interactome of PTPN22 derived from NCI-pathways (Nature 2016) using Enrichr. (B) HEK293T cells were co-transfected with SFB-PTPN22 along with Myc-tagged RICTOR, mSIN, mTOR, mLST8, RAPTOR and PRAS40. 24 h after transfection, cells were lysed in 0.3% CHAPS buffer and the lysates were immunoprecipitated with control IgG or anti-Flag antibody. The interactions were detected by immunoblotting with anti-Myc antibody. (C) Immunoprecipitation (IP) with control IgG or anti-PTPN22 antibody was performed with extracts derived from HeLa cells. Endogenous association of PTPN22 with mTOR complexes subunits (RICTOR, mTOR, and RAPTOR) were analysed by immunoblotting with specific antibodies. Due to low expression level of PTPN22 in HeLa cells, endogenous PTPN22 in input sample was shown by immunoprecipitating PTPN22 from cell extracts using its antibody. Source data are available online for this figure.

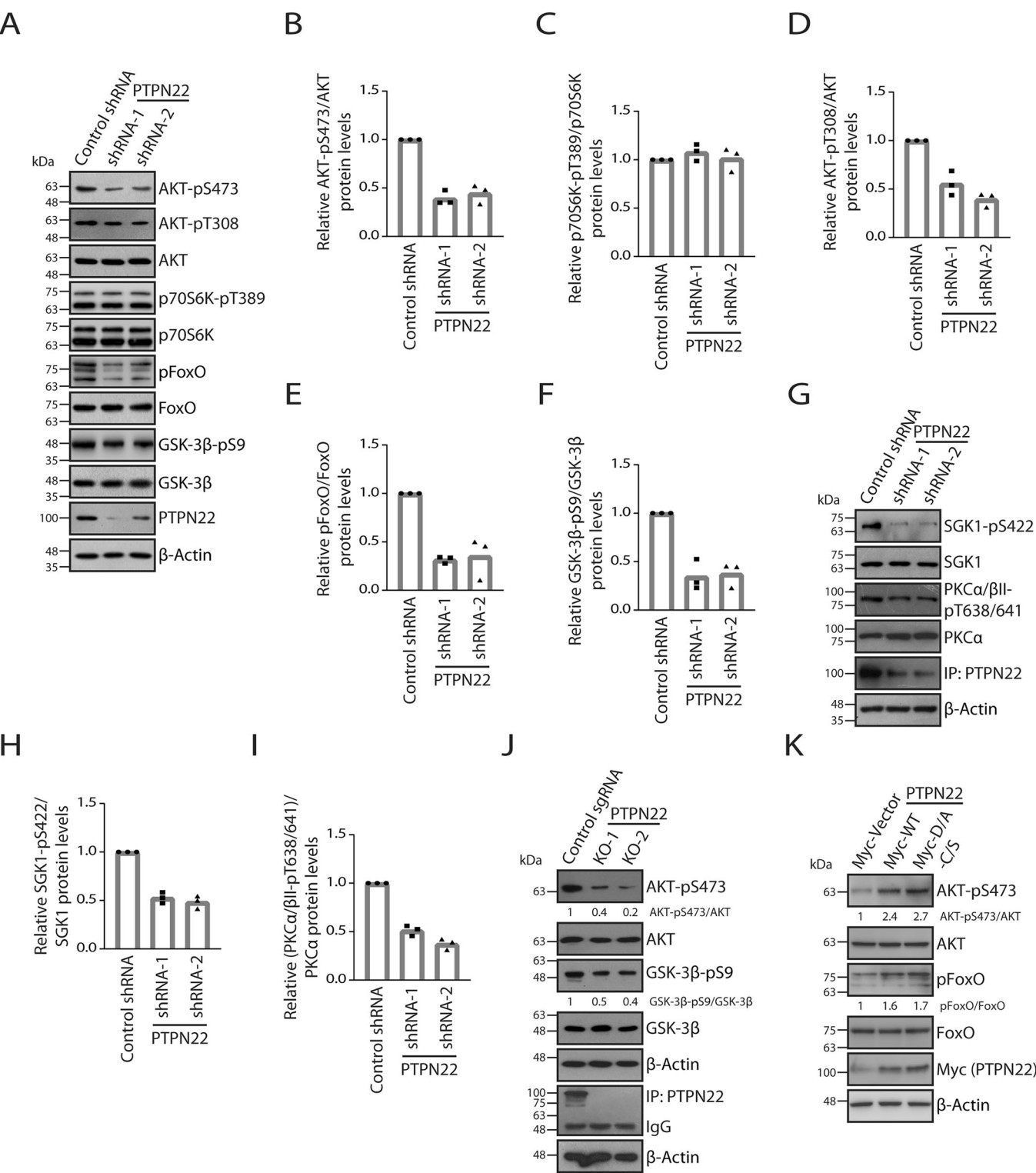

**Figure EV2. PTPN22 knockdown inhibits mTORC2 activation, related to Fig. 2.**

(A) Representative immunoblots showing analysis of whole cell lysates derived from Jurkat cells depleted of endogenous PTPN22 by two independent shRNAs (shRNA-Scramble used as a negative control) via lentiviral mediated infection, for examine the total and phosphorylated states of AKT, p70S6K, GSK-3β and FoxO with their respective antibodies. (B–F) Quantification of immunoblotting data from (A). The pixel intensity of phosphorylated protein bands were normalized by the pixel intensity of corresponding total protein bands. Individual data points from three independent experiments are shown in the graph. (G–I) Representative immunoblots showing analysis of whole cell lysates derived from HCT116 cells depleted of endogenous PTPN22 by two independent shRNAs (shRNA-Scramble used as a negative control) via lentiviral mediated infection, for examine the total and phosphorylated states of SGK1 and PKCα with their respective antibodies (G). Quantification of SGK1 phosphorylation (H) and PKCα phosphorylation (I) from (G). Individual data points from three independent experiments are shown in the graph. (J) Cell extracts of control and PTPN22 knockout cells (derived from two independent guide RNAs) were analysed by immunoblotting to determine the total and phosphorylated states of AKT and GSK-3β with specific antibodies. (K) Immunoblots showing analysis of whole cell lysates derived from Jurkat cells transfected with Myc vector, Myc-tagged PTPN22 (WT) or PTPN22 (D/A-C/S) constructs, for determine the total and phosphorylated states of AKT and FoxO with specific antibodies. Source data are available online for this figure.

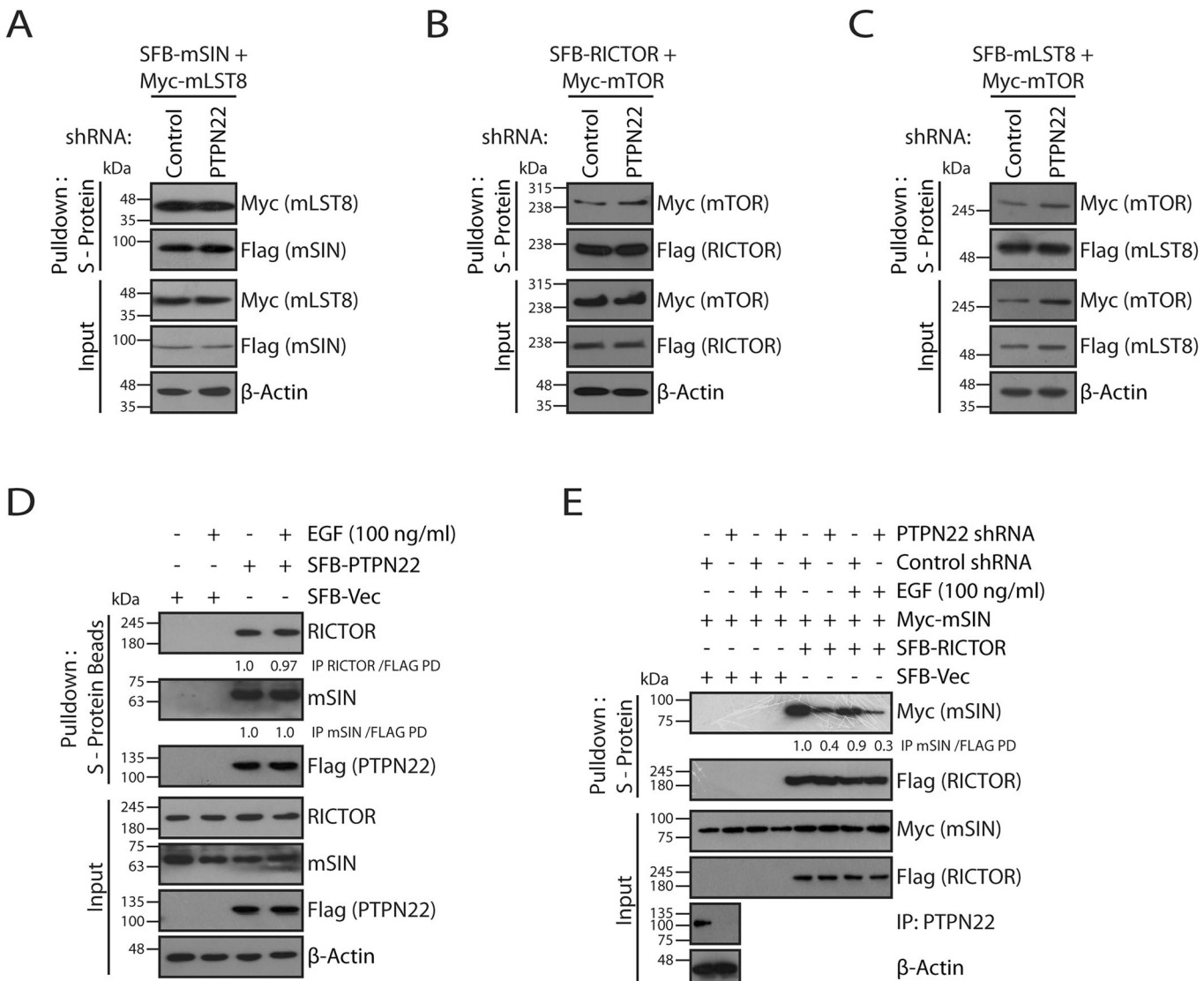

**Figure EV3. PTPN22 regulates mSIN and RICTOR association independent of growth factors, related to Fig. 3.**

(A–C) Control or PTPN22 depleted HCT116 cells were co-transfected either with (A) SFB-mSIN and Myc-mLST8, (B) SFB-RICTOR and Myc-mTOR, or (C) SFB-mLST8 and Myc-mTOR. At 48 h post-transfection, cells were lysed in 0.3% CHAPS buffer and lysates were pulldown using S-protein agarose beads. The interactions were detected by immunoblotting with anti-Myc antibody. (D) HEK293T cells were transfected with either SFB vector or SFB-PTPN22. At 24 h post-transfection, cells were serum starved for 16 h, followed by stimulation with EGF (100 ng/ml) for 15 min and lysed in 0.3% CHAPS buffer and were subjected to pulldown using S-protein agarose beads. Association of PTPN22 with RICTOR and mSIN were detected by immunoblotting with their respective antibodies. (E) Control or PTPN22 depleted HCT116 cells were co-transfected either with SFB vector or SFB-RICTOR along with Myc-mSIN. At 48 h post-transfection, cells were serum starved for 16 h, followed by stimulation with EGF (100 ng/ml) for 15 min. Cells were lysed in 1% Triton X-100 buffer and lysates were pulldown using S-protein agarose beads. The interactions were detected by immunoblotting with anti-Myc antibody. Source data are available online for this figure.

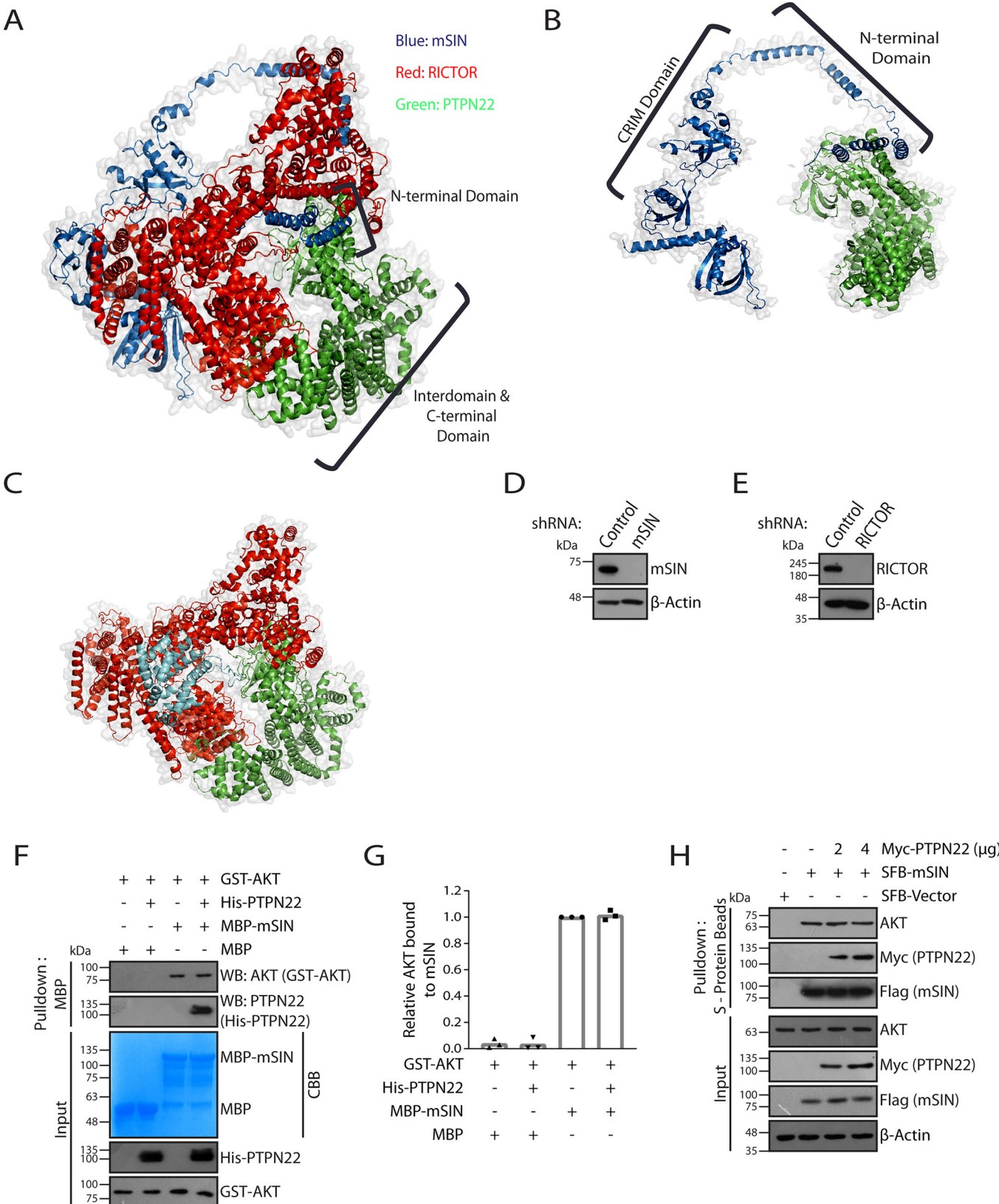

Figure EV4. **Characterization of RICTOR-mSIN-PTPN22 interacting regions, related to Fig. 4.**

(A–C) The predicted Alphafold model of RICTOR-mSIN-PTPN22 complex is shown and the proteins are highlighted in indicated colours: mSIN (*blue*), RICTOR (*red*), and PTPN22 (*green*). The N-terminus and Interdomian along with C-terminal domain of PTPN22 are denoted by the labels. ipTM = 0.49 and pTM = 0.48 represents the confidence score of the predicted structure (A). (B) 3D model depicting PTPN22 and mSIN interaction, from the predicted structure. N-terminus and the CRIM domain of mSIN are denoted by labels. (C) 3D model depicting PTPN22 and RICTOR interaction, from the predicted structure. The *cyan* colour indicates the ARM4 region of RICTOR. (D) HEK293T cells were transduced with either control shRNA or pool of multiple mSIN shRNAs, via lentiviral mediated infection. Stable cell lines were generated and knockdown efficiency was verified by immunoblotting with anti-mSIN antibody. (E) HEK293T cells were transduced with either control shRNA or pool of multiple RICTOR shRNAs, via lentiviral mediated infection, and stable cell lines were generated. The knockdown efficiency was verified by immunoblotting with anti-RICTOR antibody. (F, G) MBP or MBP-mSIN bound on dextran sepharose beads were incubated with purified GST-AKT either in the presence of recombinant PTPN22 or equal volume of corresponding buffer, and its effect on the AKT-mSIN interaction was assessed by immunoblotting MBP-pulldowns with AKT antibody (F), and individuals data points for relative AKT bound to mSIN were plotted from three independent experiments (G). (H) SFB-mSIN along with increasing concentrations (0–4 µg plasmid) of Myc-PTPN22 were transfected in HEK293T cells. At 24 h post-transfection, cells were lysed and subjected to pulldown using S-protein agarose beads, and the effect of PTPN22 on the interaction of AKT and mSIN was assessed by immunoblotting pulldowns with AKT specific antibody. SFB vector was used as a negative control. β-actin was used as a loading control. Source data are available online for this figure.

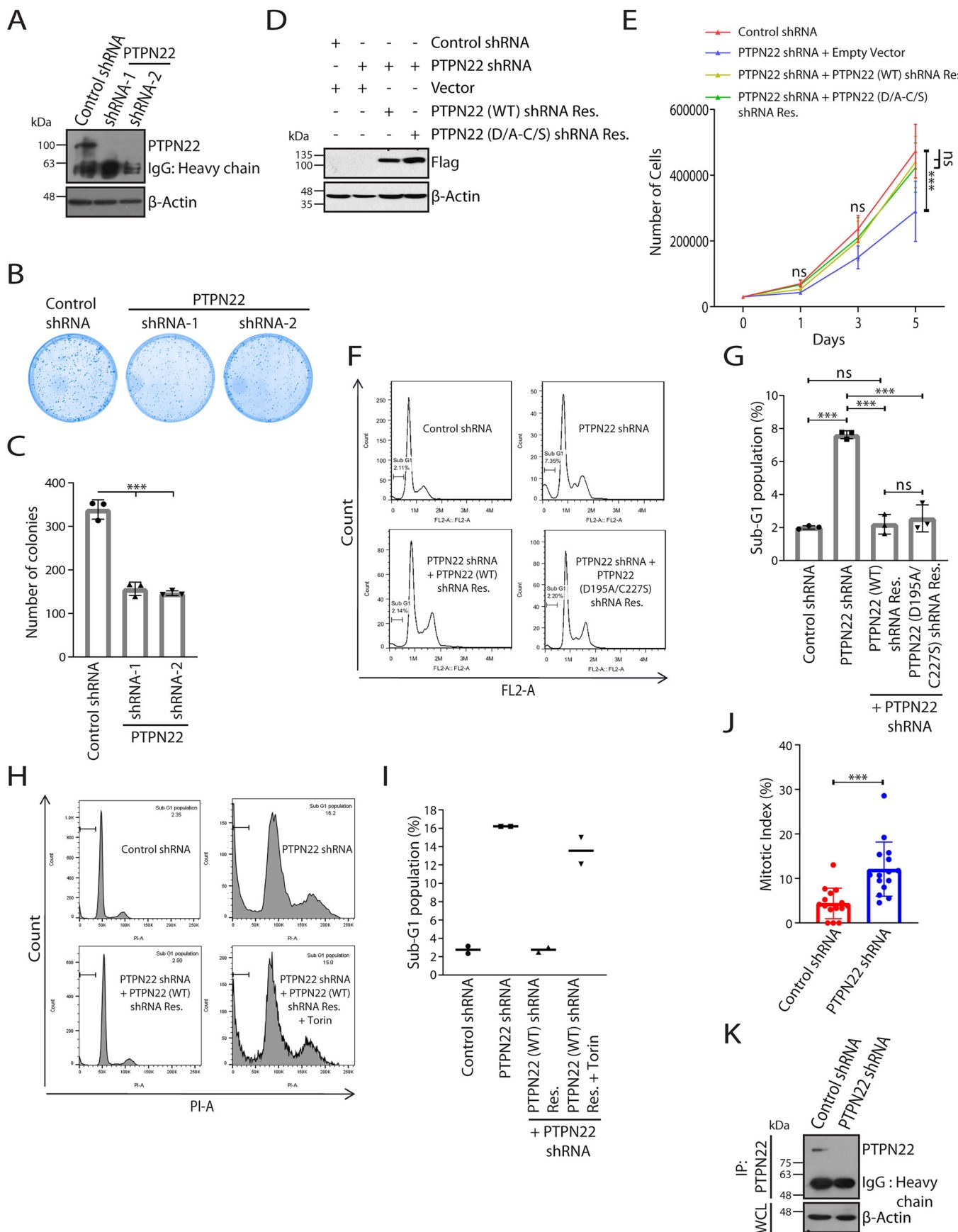

Figure EV5.  PTPN22 acts as a potential oncogene, related to Fig. 5.

(A) HCT116 cells were transduced with either control shRNA or two independent PTPN22 shRNAs via lentiviral mediated infection, and stable cell lines were generated. Knockdown efficiency was shown by immunoprecipitating PTPN22 from cell lysates using PTPN22 antibody, followed by immunoblot analysis of anti-PTPN22 immunoprecipitates and whole cell lysates with specific antibodies. (B, C) Colony formation assays were performed in HCT116 cells expressing control shRNA or PTPN22 shRNA constructs. (B) Images were captured after staining with crystal violet, and (C) quantification data for number of colonies from three independent experiments are shown. Error bars indicate mean ± standard deviation ($n = 3$), ***$P = 0.000024$ for control shRNA v/s PTPN22 shRNA-1, ***$P = 0.000018$ for control shRNA v/s PTPN22 shRNA-2 (one-way ANOVA with Bonferroni's multiple comparisons test). (D) The expression of shRNA-resistant PTPN22 wild-type (WT) or PTPN22 catalytic mutant (D/A-C/S) constructs transfected in PTPN22 depleted cells was shown by immunoblotting with anti-Flag antibody. (E) HCT116 cells expressing either control shRNA or PTPN22 shRNAs were transfected with indicated constructs. After day 1, cell number was measured every 2 days for the indicated durations. Quantification of cell number was calculated from three independent experiments. Error bars indicate mean ± SD ($n = 3$), ***$P = 0.0003$, ns: not significant (two-way ANOVA with Tukey's multiple comparisons test). (F, G) Representative images from sub-G1 population measurement by flow cytometric analysis in HCT116 cells expressing indicated constructs are shown (F), and (G) quantified data for percentage of sub-G1 population from three independent experiments were plotted. Error bars represent mean ± standard deviation ($n = 3$), ***$P = 0.00000616$ for control shRNA v/s PTPN22 shRNA, ***$P = 0.00000803$ for PTPN22 shRNA v/s PTPN22 shRNA + PTPN22 (WT) shRNA Res., ***$P = 0.0000135$ for PTPN22 shRNA v/s PTPN22 shRNA + PTPN22 (D195A/C227S) shRNA Res., ns: not significant for control shRNA v/s PTPN22 shRNA + PTPN22 (WT) shRNA Res., and for PTPN22 shRNA + PTPN22 (WT) shRNA Res. v/s PTPN22 shRNA + PTPN22 (D195A/C227S) shRNA Res. (one-way ANOVA with Bonferroni's multiple comparisons test). (H, I) Representative images from sub-G1 population measurement by flow cytometric analysis in HCT116 cells expressing indicated constructs, and were treated either with torin (250 nM) or with DMSO control for 2 h are shown (H), and scatter plot for percentage of sub-G1 population from two independent experiments were plotted (I). (J) Quantification data for the percentage of mitotic index (number of pH3 positive cells/total number of cells (DAPI) per field) in PTPN22 depleted cells as compared to the control cells were plotted. The plotted data points were from three biological replicates and represent mean ± SD, ***$P = 0.0002$ (unpaired two-tailed Student's t test). (K) Immunoblot showing PTPN22 knockdown efficiency in cell line used for nude mice xenograft experiments. Source data are available online for this figure.

