## [Peer Review File · EMBO Reports]

Phosphatase PTPN22 functions as an adaptor in the mTORC2 complex

Keshav Gupta, Nagalakshmi Kommineni, Tanuja Bogadi, Neeraja Alamuru-Yellapragada, and Subbareddy Maddika

Corresponding author(s): Subbareddy Maddika (msreddy@cdfd.org.in)

Review Timeline:

Submission Date:	30th Jan 25
Editorial Decision:	7th Mar 25
Revision Received:	2nd Jul 25
Editorial Decision:	6th Aug 25
Revision Received:	13th Aug 25
Accepted:	20th Aug 25

Editor: Deniz Senyilmaz Tiebe

Transaction Report:

Dear Reddy,

Thank you for submitting your research manuscript to our journal, which was now seen by three referees, whose reports are copied below.

Referees express interest in the proposed phosphatase independent function of PTPN22 in regulation of mTORC2. However, they also raise some concerns that need to be addressed to consider publication here.

I find the reports informed and constructive, and believe that addressing the concerns raised will significantly strengthen the manuscript. As the reports are below, and I think all points need to be addressed, I will not detail them here. Please contact me if you have questions or comments regarding the revision for further discussion (also by video chat).

Given these positive recommendations, we would like to invite you to submit a revised manuscript. Please revise your manuscript with the understanding that the referee concerns (as in their reports) must be fully addressed and their suggestions taken on board. Please address all referee concerns in a complete point-by-point response. Acceptance of the manuscript will depend on a positive outcome of a second round of review. It is EMBO reports policy to allow a single round of major experimental revision only and acceptance or rejection of the manuscript will therefore depend on the completeness of your responses included in the next, final version of the manuscript.

We realize that it is difficult to revise to a specific deadline. In the interest of protecting the conceptual advance provided by the work, we recommend a revision within 3 months. Please discuss the revision progress ahead of this time with me if you require more time to complete the revisions, or if you have questions or comments regarding the revision (also by video chat).

1. A data availability section providing access to data deposited in public databases is missing (where applicable).
2. Your manuscript contains statistics and error bars based on $n=2$. Please use scatter plots in these cases.

You can submit the revision either as a Scientific Report or as a Research Article. For Scientific Reports, the revised manuscript can contain up to 5 main figures and 5 Expanded View figures, and it should not exceed 27000 characters. If the revision leads to a manuscript with more than 5 main figures it will be published as a Research Article. In this case the Results and Discussion section should be separate. If a Scientific Report is submitted, these sections have to be combined. This will help to shorten the manuscript text by eliminating some redundancy that is inevitable when discussing the same experiments twice. In either case, all materials and methods should be included in the main manuscript file.

4) a .docx formatted letter INCLUDING the reviewers' reports and your detailed point-by-point responses to their comments. As part of the EMBO publication's Transparent Editorial Process, EMBO reports publishes online a Review Process File (RPF) to accompany accepted manuscripts. This File will be published in conjunction with your paper and will include the referee reports, your point-by-point response and all pertinent correspondence relating to the manuscript.

<https://www.embopress.org/page/journal/14693178/authorguide#transparentprocess>

5) a complete author checklist, which you can download from our author guidelines

<https://www.embopress.org/page/journal/14693178/authorguide>. Please insert information in the checklist that is also reflected in the manuscript. The completed author checklist will also be part of the RPF.

6) Please note that all corresponding authors are required to supply an ORCID ID for their name upon submission of a revised manuscript (). Please find instructions on how to link your ORCID ID to your account in our manuscript tracking system in our Author guidelines

7) Before submitting your revision, primary datasets produced in this study need to be deposited in an appropriate public database (see <https://www.embopress.org/page/journal/14693178/authorguide#datadeposition>). Please remember to provide a reviewer password if the datasets are not yet public. The accession numbers and database should be listed in a formal "Data Availability" section placed after Materials & Method (see also

<https://www.embopress.org/page/journal/14693178/authorguide#datadeposition>). Please note that the Data Availability Section is restricted to new primary data that are part of this study. * Note - All links should resolve to a page where the data can be accessed. *

Additional information on source data and instruction on how to label the files are available:

<https://www.embopress.org/page/journal/14693178/authorguide#sourcedata>

9) Our journal encourages inclusion of *data citations in the reference list* to directly cite datasets that were re-used and obtained from public databases. Data citations in the article text are distinct from normal bibliographical citations and should directly link to the database records from which the data can be accessed. In the main text, data citations are formatted as follows: "Data ref: Smith et al, 2001" or "Data ref: NCBI Sequence Read Archive PRJNA342805, 2017". In the Reference list, data citations must be labeled with "[DATASET]". A data reference must provide the database name, accession number/identifiers and a resolvable link to the landing page from which the data can be accessed at the end of the reference. Further instructions are available at <http://www.embopress.org/page/journal/14693178/authorguide#referencesformat>

10) Regarding data quantification (see Figure Legends:

<https://www.embopress.org/page/journal/14693178/authorguide#figureformat>)

- the name of the statistical test used to generate error bars and P values,
- the number (n) of independent experiments (please specify technical or biological replicates) underlying each data point,
- the nature of the bars and error bars (s.d., s.e.m.),
- If the data are obtained from n Program fragment delivered error ``Can't locate object method "less" via package "than" (perhaps you forgot to load "than"?) at //ejpvfs23/sites23b/embo_www/letters/embo_decision_revise_and_review.txt line 56.' 2, use scatter blots showing the individual data points.

12) Please also note our reference format:

13) All Materials and Methods need to be described in the main text using our 'Structured Methods' format, which is required for all research articles. According to this format, the Methods section includes a Reagents and Tools Table (listing key reagents, experimental models, software and relevant equipment and including their sources and relevant identifiers) followed by a Methods and Protocols section describing the methods using a step-by-step protocol format. The aim is to facilitate adoption of the methodologies across labs. More information on how to adhere to this format as well as a downloadable template (.docx) for the Reagents and Tools Table can be found in our author guidelines:

I look forward to seeing a revised version of your manuscript when it is ready. Please let me know if you have questions or comments regarding the revision.

Kind regards,

Deniz

Deniz Senyilmaz Tiebe, PhD
Senior Scientific Editor
EMBO Reports

Referee #1:

By analyzing the interactome of the tyrosine phosphatase PTPN22 and conducting a series of co-immunoprecipitation assays, the authors demonstrate that PTPN22 associates with mTORC2. First, they showed that PTPN22 interacts with mTORC2 subunits, but not with the mTORC1 subunit raptor. Second, they found that depletion of PTPN22 diminished phosphorylation of Akt at Ser473 and Thr308, but not the mTORC1 target p70S6K. Phosphorylation of Akt target GSK3beta and FoxO1 was also diminished under these conditions. The role of PTPN22 on Akt phosphorylation does not require its phosphatase activity as revealed by ectopic expression of a phosphatase mutant. They further examined the interaction of PTPN22 with mTORC2 and found that it enhances interaction of rictor and SIN1, specifically, and that the N- and C-terminal domains of PTPN22 are involved in interacting with these mTORC2 subunits. Lastly, knockdown of PTPN22 diminished cell number and colony formation as well as tumor formation ability of HCT116 cells in xenograft assays. Based on these findings, the authors conclude that "PTPN22 acts as an adaptor in the mTORC2 complex."

Identification of PTPN22 as an interactor of mTORC2 is interesting and adds to the growing list of regulators of mTORC2 signaling. The interaction/association studies, both using endogenous and exogenous proteins, are well-conducted. However, the function of PTPN22 in mTORC2 regulation remains obscure. Hence, the studies are highly preliminary in its current form.

Specific questions:

1. Does PTPN22 promote mTORC2 assembly (rictor-SIN1 interaction) under conditions that activate mTORC2? Is PTPN22 interaction with mTORC2 also regulated by such conditions?
2. How does conditions that activate PTPN22 affect mTORC2 assembly and/or activity?
3. The statement in the abstract: "PTPN22 is crucial for mTORC2-driven cell growth and survival..." is only speculative.
4. Since depletion of PTPN22 diminished phosphorylation of both Akt-Ser473 and -Thr308, this would indicate that it could mediate PI3K signals but there was no interrogation of this.
5. How does PTPN22 affect phosphorylation of other mTORC2 targets?

Referee #2:

The manuscript by the Maddika group provides evidence that PTPN22 interacts with mTORC2 and promotes mTORC2-Akt signaling. The data are solid, combining experiments in different cell types and analyzing both endogenous and tag-versions of the proteins. The data confirm previous studies already proposing PTPN22 as a positive regulator of Akt signaling, while adding

new evidence of a direct interaction with mTORC2. I have some minor points:

1) The effects of PTPN22 knockdown (KD) or overexpression on Akt signaling are convincing but mild. The authors should attenuate the emphasis in the description of some figures. For instance, the EGF-mediated stimulation of Akt phosphorylation is not completely abrogated by PTPN22 KD, as a residual signal is clearly detected (Figure 2D). The same in other cell types (Figure 2C, EV2A). It is possible that the residual Akt phosphorylation may also be due to an inefficient KD, as PTPN22 mRNA and protein are still detected after shRNA (Fig. 2A, 2C). A KO line by genome editing could have been helpful to conclude about this point.

2) The authors cannot really conclude about the effects of PTPN22 on cell proliferation as they did not measure cell cycle progression, S-phase entry or mitosis but just counted cell number. The mild difference in cell number after PTPN22 KD shown in Figure 5A could be due to the induction of apoptosis. Is the induction of apoptosis also triggered by Torin treatment? Or is it due to an mTOR independent effect of PTPN22?

Referee #3:

EMBO Reports Review_Gupta 2025

In this interesting and well conducted piece of work, the authors show that the protein phosphatase PTPN22 can interact with the mTORC2 complex and affect its ability to activate one of its key client kinases Akt. Authors show interaction of endogenous PTPN22 specifically with mTORC2 and this is convincing. Attempts to recapitulate this as much as possible with recombinant proteins are also made which is useful. Careful work is conducted to map the interactions between PTPN22 and the mTORC2 specific components Sin1 and Rictor which is also to be commended. The authors further indicate that this interaction promotes mTORC2 function both in vitro and in an subcut-tumour model. The paper is useful and adds another regulatory component to the mTORC2 story. There are however a few issues which could be addressed, particularly with regard to consistency in results display, quantitation and interpretation. Some specific pointers are provided below which might improve the work further.

Why in panel 1G is the critical PTPN2 blot split? In legend it explains this is because "endogenous PTPN22 are low in HEK293T cells". The blot should still be shown in full for this section and an IP showing PTPN22 could be shown in addition below. As displayed it feels a bit misleading.

Why are there two input lysates shown for panel C (IgG and mSin1) - were different lysates used? The different experimental layout for B and C is a little confusing.

1F shows expression of PTPN22 by RT-PCR in a small but diverse panel. Why were Western blots not used for this also? As presented this could be more convincing. There is plenty of publicly available data on gene expression in cell lines, tissues and disease for all genes.

In figure 1 size exclusion chromatography, Rictor overlap with Sin and PTPN22 is only partial. Could the authors comment on this? This data is nonetheless convincing.

Figure 2 - changes to AKT phos need to be quantified from multiple biological replicates and presented as graphs. This includes the majority of panels in Figure 2. The changes to EGF induced Akt phosphorylation look convincing but the quality of the blot images in some look poor (scan quality?) - 2D and 2E are good examples as they are not as high quality as many others (milky).

In Figure 3B, PTPN22 appears to precipitate in the IgG control?

Data in Fig 3 is shown that suggest interaction with both Sin1 and Rictor and that PTPN22 might act as a scaffold to support Sin1 and Rictor association. For panel 3C, Rictor, immune precipitated from Sin1 depleted cells, is shown to bind recombinant PTPN22. This is perhaps not entirely convincing unless Sin1 is knocked out rather than depleted. Could these blots be probed for Sin1 to better demonstrate that it is absent from complexes? It is also proposed that PTPN22 regulates Sin1-Rictor complex stability. In these experiments using shRNA, PTPN22 blots really need to be included in the main figure. Perhaps CRISPR-Cas9 could be used to delete PTPN22 as an orthogonal approach? The recapitulation of interactions in vitro is to be commended however.

In pull downs, there is inconsistency in whether inputs are displayed or not. Could the authors either include or clarify why they are sometimes in and sometimes out? Data 3F, 3G, 3H.

The authors show in panel 4A that the CRIM domain is key for interaction and panel 4A suggests this will be with the catalytic domain of PTP22. As this fairly small ubiquitin fold domain has been variously reported to directly recruit mTORC2 substrates (PMID: 28264193; 21806543), how does PTPN22 interaction with this domain impact on substrate/client kinase association with the complex? It would also be worth examining the results in context of the multiple Cryo-EM structures reported for the whole mTORC2 complex (e.g. PMID: 33158864; 29424687; 29567957 and Rictor/Sin substrate interaction structural studies (e.g.

PMID: 35926713). This at least s requires careful discussion.

Interaction with Rictor via the ARM4 region also looks convincing. Given the fairly well mapped interaction domains, might there be space for modelling interactions between PTPN22, Sin1 and/or Rictor using Alphafold?

Consistent with the suppression of Akt phosphorylation, suppression of PTPN22 results in reduced cell growth both in vitro and in a subcut tumour model. For this type of study, these validations are useful and sufficient.

Point-by-point response to reviewers

Referee #1:

By analyzing the interactome of the tyrosine phosphatase PTPN22 and conducting a series of co-immunoprecipitation assays, the authors demonstrate that PTPN22 associates with mTORC2. First, they showed that PTPN22 interacts with mTORC2 subunits, but not with the mTORC1 subunit raptor. Second, they found that depletion of PTPN22 diminished phosphorylation of Akt at Ser473 and Thr308, but not the mTORC1 target p70S6K. Phosphorylation of Akt target GSK3beta and FoxO1 was also diminished under these conditions. The role of PTPN22 on Akt phosphorylation does not require its phosphatase activity as revealed by ectopic expression of a phosphatase mutant. They further examined the interaction of PTPN22 with mTORC2 and found that it enhances interaction of rictor and SIN1, specifically, and that the N- and C-terminal domains of PTPN22 are involved in interacting with these mTORC2 subunits. Lastly, knockdown of PTPN22 diminished cell number and colony formation as well as tumor formation ability of HCT116 cells in xenograft assays. Based on these findings, the authors conclude that “PTPN22 acts as an adaptor in the mTORC2 complex”.

Response: We thank the reviewer for excellent summary of our work.

Identification of PTPN22 as an interactor of mTORC2 is interesting and adds to the growing list of regulators of mTORC2 signaling. The interaction/association studies, both using endogenous and exogenous proteins, are well-conducted. However, the function of PTPN22 in mTORC2 regulation remains obscure. Hence, the studies are highly preliminary in its current form.

Response: We thank the reviewer for appreciating our study, acknowledging the novelty of the study and for highlighting the important aspects of the manuscript. The constructive suggestions are greatly helpful to improve the impact of our manuscript. In the revised manuscript, we have provided several additional sets of data in relation to the function of PTPN22 in mTORC2 regulation (point by point response below).

Specific questions:

1. Does PTPN22 promote mTORC2 assembly (rictor-SIN1 interaction) under conditions that activate mTORC2? Is PTPN22 interaction with mTORC2 also regulated by such conditions?

Response: As per the reviewer's suggestion, we have now tested the association of RICTOR-mSIN1 in response to epidermal growth factor (mTORC2 activator) or serum-starvation in PTPN22 knockdown cells in comparison to control cells. Notably, we found PTPN22 is required for RICTOR-mSIN1 association also under conditions that activate mTORC2 (revised figure EV3E); suggesting PTPN22 promotes RICTOR-mSIN1 association,

irrespective of the availability of growth factors. Moreover, in order to establish the role of growth factors in PTPN22-mTORC2 interaction, we performed co-immunoprecipitation of SFB-PTPN22 with RICTOR and mSIN1, with or without addition of growth factor (EGF). As shown in the revised figure EV3D, the interaction of PTPN22 with mSIN and RICTOR remains unchanged, suggesting that PTPN22 associates with mTORC2 components in growth factors-independent manner.

2. How does conditions that activate PTPN22 affect mTORC2 assembly and/or activity?

Response: In line with this suggestion, treatment of cells with phorbol 12-myristate 13-acetate (PMA), which leads to an increase in the amount of PTPN22 protein (refer to PMID:22569400, PMID:32184287, PMID:19299707), results in increased interaction between RICTOR and mSIN (revised figure 3J), as more PTPN22 protein is available to promote mSIN and RICTOR assembly. In addition to this, we found treatment with PMA promotes the phosphorylation of AKT-pS473, however this increased phosphorylation of AKT is not due to phosphatase activity of PTPN22, as treatment of cells with PTPN22 inhibitor, (PTPN22-IN-1) does not affect AKT phosphorylation at Ser473. Moreover, treatment with PTPN22-IN-1 is not able to rescue the increased AKT-pS473 phosphorylation by PMA (revised figure 3K); suggesting that the PTPN22 protein levels, but not its catalytical activity, is required for mTORC2 mediated AKT activation. Our newly generated data support our conclusion that PTPN22 functions as an adaptor in the mTORC2 complex.

3. The statement in the abstract: “PTPN22 is crucial for mTORC2-driven cell growth and survival...” is only speculative.

Response: As per the reviewer’s suggestion, the statement has been appropriately modified in the revised manuscript.

4. Since depletion of PTPN22 diminished phosphorylation of both Akt-Ser473 and -Thr308, this would indicate that it could mediate PI3K signals but there was no interrogation of this.

Response: As noted by the reviewer, depletion of PTPN22 diminished both Ser-473 and Thr-308 phosphorylation of AKT. It is well known in the literature that the status of AKT-pT308 phosphorylation by PDK1 is dependent upon prior phosphorylation of AKT-pS473 by mTORC2 (refer to PMID:15718470, PMID:12167717, PMID:12086620). Therefore the reduction seen in the AKT-pT308 phosphorylation can be attributed to this indirect regulation through mTORC2. Earlier, Bai et al. reported that PTPN22 can activate the PI3K signaling via 14-3-3 τ adapter protein in T cells (PMID:37255287). Also, Liang et al. recently reported that PTPN22 can modulate mitophagy and pyroptosis in human and rat degenerative nucleus pulposus (NP) cells through the regulation of PI3K/AKT/mTOR axis by inhibited the phosphorylation of PI3K (p85 Tyr458), AKT (Ser473) and mTOR (Ser2448) via its interaction with PI3K (p110 α) (PMID:40349959). However, in our mass-spectrometry based interactome studies of PTPN22, we did not find any PI3K proteins or its interacting partners, therefore we did not further focus on this angle in this study.

5. How does PTPN22 affect phosphorylation of other mTORC2 targets?

Response: In line with this suggestion, we now tested the effect of PTPN22 depletion with respect to other mTORC2 targets; SGK1 and PKC α phosphorylation. As seen in revised figure EV2G-EV2I, we found SGK1-pS422 and phospho-PKC α / β II [pT638/641] phosphorylation were reduced in PTPN22 knockdown cells in comparison to control cells, possibly suggesting that the effects of PTPN22 can be generalized to multiple mTORC2 targets.

Referee #2:

The manuscript by the Maddika group provides evidence that PTPN22 interacts with mTORC2 and promotes mTORC2-Akt signaling. The data are solid, combining experiments in different cell types and analyzing both endogenous and tag-versions of the proteins. The data confirm previous studies already proposing PTPN22 as a positive regulator of Akt signaling, while adding new evidence of a direct interaction with mTORC2. I have some minor points:

Response: We thank the reviewer for recognizing that our data is solid, acknowledging the novelty of the study in connection with previously published studies and for highlighting the important aspects of the manuscript. The constructive suggestions are greatly helpful to improve the impact of our manuscript. We have addressed your concerns point-by-point below.

1) The effects of PTPN22 knockdown (KD) or overexpression on Akt signaling are convincing but mild. The authors should attenuate the emphasis in the description of some figures. For instance, the EGF-mediated stimulation of Akt phosphorylation is not completely abrogated by PTPN22 KD, as a residual signal is clearly detected (Figure 2D). The same in other cell types (Figure 2C, EV2A). It is possible that the residual Akt phosphorylation may also be due to an inefficient KD, as PTPN22 mRNA and protein are still detected after shRNA (Fig. 2A, 2C). A KO line by genome editing could have been helpful to conclude about this point.

Response: We agree with the reviewer that the AKT phosphorylation is not completely abrogated by PTPN22 knockdown. Therefore, as per the reviewer's suggestion, the text has been appropriately modified in the revised manuscript and the word "significantly reduced" has been changed to "reduced" in the description of figure EV2A. Similarly, the word "abrogated" has been changed to "impaired" in the description of original figure 2D (now revised figure 2E), and the word "strongly suppressed" has been changed to "suppressed" in the description of original figure 2H (now revised figure 2L). Additionally, as suggested by the reviewer, we generated PTPN22 knockout cell lines using two independent CRISPR-Cas9 based guide RNAs, and analysed the phosphorylation status of AKT-pS473 and its downstream target GSK-3 β -pS9 in comparison to wild type cells. As seen in revised figure EV2J, PTPN22 KO cells displayed reduced phosphorylation of AKT and GSK-3 β as compared to control cells. Moreover, the extent of reduction in phosphorylation of AKT and

GSK-3 β are stronger in case of PTPN22 KO cells as compared to PTPN22 knockdown cells (comparison between figure EV2J and 2C).

2) The authors cannot really conclude about the effects of PTPN22 on cell proliferation as they did not measure cell cycle progression, S-phase entry or mitosis but just counted cell number. The mild difference in cell number after PTPN22 KD shown in Figure 5A could be due to the induction of apoptosis.

Response: To address the reviewer's concern, we now assessed the mitotic index between PTPN22 knockdown cells in comparison to control cells by phospho-histone H3 (pH3) staining. As seen in revised figure EV5J, the mitotic index (pH3/DAPI) was significantly increased in PTPN22 knockdown cells in comparison to control shRNA cells, possibly suggesting a delay in cell cycle progression. Moreover, at several places in the manuscript we have replaced the word "cell proliferation" with "cell number", and used the word cell proliferation wherever it is appropriately justified with experimental evidence.

Is the induction of apoptosis also triggered by Torin treatment? Or is it due to an mTOR independent effect of PTPN22?

Response: We thank the reviewer for highlighting this point. As, we previously demonstrated that the knockdown of PTPN22 leads to increased apoptosis, (as measured by increase in subG1 population of cells), which can be reduced back similar to control cells by expressing PTPN22 wild-type or catalytically dead mutant (refer to figure EV5F and EV5G). In order to address the reviewer's concern, we treated PTPN22 overexpressing cells (PTPN22 overexpressed in the depletion background of PTPN22) with Torin1. As seen in revised figures EV5H and EV5I, Torin treatment prevents the rescue of apoptosis by PTPN22 overexpression in PTPN22 depleted cells, suggesting that the apoptotic effects of PTPN22 depletion are mediated through mTORC2 activation

Referee #3:

EMBO Reports Review_Gupta 2025

In this interesting and well conducted piece of work, the authors show that the protein phosphatase PTPN22 can interact with the mTORC2 complex and affect its ability to activate one of its key client kinases Akt. Authors show interaction of endogenous PTPN22 specifically with mTORC2 and this is convincing. Attempts to recapitulate this as much as possible with recombinant proteins are also made which is useful. Careful work is conducted to map the interactions between PTPN22 and the mTORC2 specific components Sin1 and Rictor which is also to be commended. The authors further indicate that this interaction promotes mTORC2 function both in vitro and in an subcut-tumour model. The paper is useful and adds another regulatory component to the mTORC2 story. There are however a few issues which could be addressed, particularly with regard to consistency in results display, quantitation and interpretation. Some specific pointers are provided below which might improve the work further.

Response: We thank the reviewer for appreciating our study, acknowledging the novelty of the study and for highlighting the important aspects of the manuscript. The constructive suggestions are greatly helpful to improve the impact of our manuscript. We have addressed your concerns point-by-point below.

Why in panel 1G is the critical PTPN22 blot split? In legend it explains this is because “endogenous PTPN22 are low in HEK293T cells”. The blot should still be shown in full for this section and an IP showing PTPN22 could be shown in addition below. As displayed it feels a bit misleading.

Response: As per the reviewer’s suggestion we have now incorporated the full blot for both IP and input panels in the original figure 1G (now revised figure 1H).

Why are there two input lysates shown for panel C (IgG and mSin1)-were different lysates used? The different experimental layout for B and C is a little confusing.

Response: The same cell lysate was loaded into two different lanes (input) shown for panel 1C (for IgG and mSIN1). However, as per the reviewer’s suggestion to avoid confusion and maintain uniformity we have cropped one of the input lanes, as the second lane represents the same lysate as mentioned.

1F shows expression of PTPN22 by RT-PCR in a small but diverse panel. Why were Western blots not used for this also? As presented this could be more convincing. There is plenty of publicly available data on gene expression in cell lines, tissues and diseases for all genes.

Response: As suggested by the reviewer, Western blots showing expression of PTPN22 in the same panel of cell lines have now been included (revised figure 1G).

In figure 1 size exclusion chromatography, Rictor overlap with Sin and PTPN22 is only partial. Could the authors comment on this? This data is nonetheless convincing.

Response: Non-overlapping low molecular weight fractions of mSIN and PTPN22 corresponds to either monomeric form or may be protein bound to other associated interactors, as these proteins are known to function independent of the mTORC2 pathway, [for mSIN; e.g. PMID:23727834, PMID:17054722], [for PTPN22; e.g. PMID:39173071, PMID:16461343, PMID:32184287]. Moreover, it is well accepted in mTORC2 literature that during size exclusion chromatography, a fraction of mSIN appears to be eluted in low molecular weight fractions (PMID:28489822, PMID:24161930).

Figure 2- changes to AKT phos need to be quantified from multiple biological replicates and presented as graphs. This includes the majority of panels in Figure 2.

Response: We thank the reviewer for highlighting this point. All the experiments in figure 2, wherever changes in AKT phosphorylation were measured have been performed in three biological replicates and presented as graphs, which have now been included in revised figure 2 (panel 2D, 2F, 2H, 2J and 2M).

The changes to EGF induced Akt phosphorylation look convincing but the quality of the blot images in some look poor (scan quality?) - 2D and 2E are good examples as they are not as high quality as many others (milky).

Response: As per the reviewer's suggestion we have replaced the blots of panel 2D and 2E (original figure) with high quality blots (now revised figure 2E and 2G).

In Figure 3B, PTPN22 appears to precipitate in the IgG control?

Response: We thank the reviewer for highlighting this point; it is due to non-specific binding of PTPN22 with protein G beads. However, we repeated this experiment with stringent washing and replaced the image panel with better representative blots in the revised manuscript (figure 3B).

Data in Fig 3 is shown that suggest interaction with both Sin1 and Rictor and that PTPN22 might act as a scaffold to support Sin1 and Rictor association. For panel 3C, Rictor immune precipitated from Sin1 depleted cells, is shown to bind recombinant PTPN22. This is perhaps not entirely convincing unless Sin1 is knocked out rather than depleted. Could these blots be probed for Sin1 to better demonstrate that it is absent from complexes?

Response: We agree with the reviewer that the PTPN22 direct binding with RICTOR immunoprecipitated for SIN1 depleted cells may not be fully convincing. Therefore, as per the reviewer's suggestion, we tried generating mSIN1 knockout cell lines using two independent guide RNAs. Unfortunately, the generated clones are having incomplete loss of mSIN1, as clones having complete knockout of mSIN1 are slow growing (due to reduction of mTORC2 signaling), and may be underrepresented in the clonal population after selection. However, to rule out the involvement of mSIN1 in PTPN22-RICTOR association, we now performed in vitro binding experiments using purified MBP-PTPN22 and bacterial cell lysates expressing GST-RICTOR. This data clearly suggests a direct interaction between PTPN22 and RICTOR (now revised figure 3C).

It is also proposed that PTPN22 regulates Sin1-Rictor complex stability. In these experiments using shRNA, PTPN22 blots really need to be included in the main figure. Perhaps CRISPR-Cas9 could be used to delete PTPN22 as an orthogonal approach? The recapitulation of interactions in vitro is to be commended however.

Response: As per the reviewer's suggestion we have now incorporated the PTPN22 gel showing knockdown by shRNA in the main figure (now revised figure 3E). In addition, as suggested by the reviewer, we generated PTPN22 knockout cell lines using CRISPR-Cas9 based guide RNAs, and analysed the mSIN-RICTOR complex stability. Consistent with our previous findings, knockout of PTPN22 diminished mSIN-RICTOR association (revised figure 3G), suggesting the potential adaptor role of PTPN22 in mSIN-RICTOR assembly. Additionally, PTPN22 knockout cells displayed a significant reduction in AKT-pS473 phosphorylation and its downstream target GSK-3 β -pS9 phosphorylation (revised figure EV2J), [Reviewer # 2 concern; point 1]. Moreover, we thank the reviewer for appreciating our in vitro data.

In pull downs, there is inconsistency in whether inputs are displayed or not. Could the authors either include or clarify why they are sometimes in and sometimes out? Data 3F, 3G, 3H.

Response: We made the suggested correction. Inputs are displayed uniformly in the revised figure (revised figure EV3A, EV3B and EV3C) to maintain uniformity.

The authors show in panel 4A that the CRIM domain is key for interaction and panel 4A suggests this will be with the catalytic domain of PTPN22. As this fairly small ubiquitin fold domain has been variously reported to directly recruit mTORC2 substrates (PMID: 28264193; 21806543), how does PTPN22 interaction with this domain impact on substrate/client kinase association with the complex? It would also be worth examining the results in context of the multiple Cryo-EM structures reported for the whole mTORC2 complex (e.g. PMID: 33158864; 29424687; 29567957) and Rictor/Sin substrate interaction structural studies (e.g. PMID: 35926713). This at least s requires careful discussion.

Response: As per the reviewer's suggestion to test the effect of PTPN22 interaction with mSIN-CRIM domain on its substrate (essential for the recruitment of mTORC2 substrates), we performed in vitro competition experiments using bacterially purified proteins which demonstrate that the binding of mSIN with PTPN22 did not affect the binding of AKT (mTORC2 substrate) with mSIN (revised figures EV4F and EV4G). Moreover, in cells the interaction of AKT with mSIN remains unchanged with increasing levels of PTPN22 (revised figures EV4H), possibly suggesting that the binding interfaces of mSIN-CRIM domain are different for AKT and PTPN22. Moreover, as per the reviewer's suggestion, the appropriate discussion has been included in the revised manuscript in context with the available mTORC2 complex structures and our domain mapping experiments.

Interaction with Rictor via the ARM4 region also looks convincing. Given the fairly well mapped interaction domains, might there be space for modelling interactions between PTPN22, Sin1 and/or Rictor using Alphafold?

Response: As suggested by the reviewer, we have modelled the interactions between PTPN22, SIN1 and RICTOR using Alphafold server. Consistent with our biochemical data, predicted structure displayed an interaction of mSIN with the N-terminal region of PTPN22, while the C-terminal proline rich domain of PTPN22 is mediating its interaction with RICTOR (revised figures EV4A). The predicted structure revealed an interaction of N-terminus of mSIN and with the N-terminal catalytic domain of PTPN22, which is consistent with our biochemical data (revised figures EV4B). Although, we cannot exclude the possibility of the involvement of CRIM domain, collectively our observations indicate that the mSIN association with PTPN22, is largely through N-terminus. In addition to this, although alphafold predicted model did not align direct association of ARM4 region of RICTOR with PTPN22 (revised figures EV4C), it predicted an interaction near C-terminal domain of RICTOR with C-terminal domain of PTPN22. This slight discordance in the predicted model may be due to differences in the alignment of ARM4 region in the predicted models vs the previously determined cryo-EM mTORC2 structures (PMID:35926713, PMID:33158864). Nonetheless,

our biochemical experiments strongly suggested that ARM4 region of RICTOR is sufficient to interact with PTPN22.

Consistent with the suppression of Akt phosphorylation, suppression of PTPN22 results in reduced cell growth both in vitro and in a subcut tumour model. For this type of study, these validations are useful and sufficient.

Response: We thank the reviewer for highlighting the important aspects of the manuscript and appreciating our functional data.

Dear Reddy,

Thank you for submitting your revised manuscript. It has now been seen by all of the original referees.

As you will see, referees find that the study is significantly improved during revision and recommend publication. However, referee #3 has a few minor outstanding concerns. Please address these concerns and provide a point-by-point response. Please contact me if you would like to discuss any of these points further.

Moreover, the editorial points below need to be addressed before I can accept the manuscript.

- Please reduce the number of keywords to 5.
- The Data Availability section needs to be moved before the Acknowledgements section and the sentence in the section needs to be replaced with the following: This study includes no data deposited in external repositories.
- Please remove the Author Contributions section from the manuscript text.
- We note the following regarding source data: all source data need to be uploaded as one zipped folder per figure (not just Figure 2, 3 and 5). Inside each zipped folder, we need one file/subfolder per panel.
- As per our policy, funding information should be complete both in the manuscript text and the manuscript submission system. We note that funding information regarding 'the Council of Scientific and Industrial Research (CSIR), India' is currently missing from the manuscript submission system.
- The synopsis image needs to be 550px wide and 300-600px high. Your synopsis image is currently too tall. Please re-submit a synopsis image in the mentioned size. You can consider depicting the signaling axis horizontally instead of vertically.
- Please remove the Reagents & Tool table from the manuscript and submit it as a separate word file by using the relevant file type in the manuscript submission system.
- Please rename the "Materials and Methods" section as "Methods".
- We note that the emails sent to Neeraja P Alamuru-Yellapragada at pavanineeraja@cdfd.org.in bounces back. Please either remove the author and then re-add with a new email address or send us the new email and we will update the account on our end.
- Our production/data editors have asked you to clarify several points in the figure legends - Figure Legends (main + EV):
 - o Please note that the exact p values are not provided in the legends of figures 2M, 5C, EV5 C, G.
 - o Please note that the measure of center for the error bars needs to be defined in the legends of figures 3I, EV5 C, E, G.

Thank you again for giving us to consider your manuscript for EMBO Reports, I look forward to your minor revision.

Kind regards,

Deniz

--

Deniz Senyilmaz Tiebe, PhD
Senior Scientific Editor
EMBO Reports

Referee #1:

The authors have satisfactorily addressed this reviewer's questions and comments.

Referee #2:

The authors addressed the previously raised issues.

Referee #3:

I am happy with all the revisions made but just have a few comments with regard to editorial policy. I appreciate the efforts made to address the majority of issues raised.

The efforts to quantify all the Akt data are massively appreciated and improve the work considerably. It is noted however that all quant data are normalised to the control condition (to equal 1). This method of quantitation is slightly tricky statistically as essentially it sets the SD at zero for the control (which means applying t-test or ANOVA statistics is not formally valid). I would advise normalising data for each experiment to the sum of the dataset rather than to the control condition to avoid this statistical

trap. When compiled from replicates, the data can be scaled to make the control average = 1 but it just means you have error for all data points). I leave this decision to the EMBO Repts editorial team.

2D should really be two graphs as they are separate data (this may become needed if above point is addressed).

2J - a stats bar comparing the first two bars might also be included.

Figures 2 and 3 have somewhat strange panel orders. This is a bit confusing to follow.

Q. Why is PTPN2 quant shown as RT-PCR (3E) rather than as a Western blot under panel 3F? The additional CRISPR data solidify the conclusions so this is not a huge issue. Just seems a little odd.

Point-by-point response

Editorial points:

- Please reduce the number of keywords to 5.

Response: We have reduced the number of keywords to 5 in the revised manuscript.

- The Data Availability section needs to be moved before the Acknowledgements section and the sentence in the section needs to be replaced with the following: This study includes no data deposited in external repositories.

Response: The suggested changes have been done in the revised manuscript.

- Please remove the Author Contributions section from the manuscript text.

Response: We have removed the Author Contributions section from the revised manuscript text.

- We note the following regarding source data: all source data need to be uploaded as one zipped folder per figure (not just Figure 2, 3 and 5). Inside each zipped folder, we need one file/subfolder per panel.

Response: The source data has been changed accordingly (as per instructions).

- As per our policy, funding information should be complete both in the manuscript text and the manuscript submission system. We note that funding information regarding 'the Council of Scientific and Industrial Research (CSIR), India' is currently missing from the manuscript submission system.

Response: We have updated the funding information regarding 'the Council of Scientific and Industrial Research (CSIR), India' in the manuscript submission system.

- The synopsis image needs to be 550px wide and 300-600px high. Your synopsis image is currently too tall. Please re-submit a synopsis image in the mentioned size. You can consider depicting the signaling axis horizontally instead of vertically.

Response: The synopsis image has been prepared as per instructions, and re-submitted.

- Please remove the Reagents & Tool table from the manuscript and submit it as a separate word file by using the relevant file type in the manuscript submission system.

Response: As suggested, we have now submitted the Reagents & Tool table as a separate word file.

- Please rename the "Materials and Methods" section as "Methods".

Response: As suggested, we have now renamed the "Materials and Methods" section as "Methods".

- We note that the emails sent to Neeraja P Alamuru-Yellapragada at pavanineeraja@cfd.org.in bounces back. Please either remove the author and then re-add with a new email address or send us the new email and we will update the account on our end.

Response: The updated email address of the Neeraja P Alamuru-Yellapragada (ORCID ID: 0009-0000-6060-2570) is nrneeraja479@gmail.com

- Our production/data editors have asked you to clarify several points in the figure legends-Figure Legends (main + EV):
 - Please note that the exact p values are not provided in the legends of figures 2M, 5C, EV5 C, G.
 - Please note that the measure of center for the error bars needs to be defined in the legends of figures 3I, EV5 C, E, G.

Response: The exact p values and the measure of center (mean) have been mentioned in the figure legends of respective figures in the revised manuscript.

Referees comments:

Referee #1:

The authors have satisfactorily addressed this reviewer's questions and comments.

Response: We thank the reviewer for appreciating our revision work.

Referee #2:

The authors addressed the previously raised issues.

Response: We thank the reviewer for appreciating our revision work.

Referee #3:

I am happy with all the revisions made but just have a few comments with regard to editorial policy. I appreciate the efforts made to address the majority of issues raised.

Response: We sincerely thank the reviewer for appreciating our revision work.

The efforts to quantify all the Akt data are massively appreciated and improve the work considerably. It is noted however that all quant data are normalised to the control condition (to equal 1). This method of quantitation is slightly tricky statistically as essentially it sets the SD at zero for the control (which means applying t-test or ANOVA statistics is not formally valid). I would advise normalising data for each experiment to the sum of the dataset rather than to the control condition to avoid this statistical trap. When compiled from replicates, the data can be scaled to make the control average = 1 but it just means you have error for all data points). I leave this decision to the EMBO Reps editorial team.

Response: We thank the reviewer for the valuable comment regarding our quantification of the western data. As requested by the reviewers, we quantified all the data and presented after normalising the data of each experiment with respective control in that particular experiment. We agree that normalizing to the control condition mathematically sets its variability to zero in the control dataset. However, this approach is a widely accepted and standard practice in the field for presenting immunoblot quantification data, as it allows direct comparison of relative changes across multiple experiments. Given that this method adheres to standard reporting conventions across journals including EMBO Reports, we believe that the current representation is appropriate and request that it be accepted in its present form. Further, after discussing with the editorial team, we removed the statistics/p values applied in the graphs presented in the following panels with graphs kept as is (where individual data points are represented in the histograms): Fig 2D, F, H, H, J, M, Fig 3I, EV Fig 2B, C, D, E, F, H, I and EV4G.

2D should really be two graphs as they are separate data (this may become needed if above point is addressed).

Response: Since data normalized is set to 1 in control as mentioned in the response to the previous point, the data in 2D remains unchanged.

2J-a stats bar comparing the first two bars might also be included.

Response: We have included the relevant information in the figure 2J in the revised manuscript.

Figures 2 and 3 have somewhat strange panel orders. This is a bit confusing to follow.

Response: We have changed the panel orders for figure 2 and 3 in the revised manuscript.

Q. Why is PTPN22 quant shown as RT-PCR (3E) rather than as a Western blot under panel 3F? The additional CRISPR data solidify the conclusions so this is not a huge issue. Just seems a little odd.

Response: The panel 3E showing PTPN22 levels in knockdown v/s control cells are stable cell lines, in which we have performed experiments representing in figure 3F, as well as figure EV3A-EV3C, therefore shown separately rather than under panel 3F.

Dr. Subbareddy Maddika
Centre for DNA Fingerprinting & Diagnostics
Laboratory of Cell Death & Cell Survival
Uppal
Hyderabad, Telangana 500039
India

Dear Reddy,

Thank you for submitting your revised manuscript and swiftly addressing the outstanding minor points. I have now looked at everything and all is fine. Therefore, I am very pleased to accept your manuscript for publication in EMBO Reports.

Congratulations on a nice work!

Kind regards,

Deniz

--

Deniz Senyilmaz Tiebe, PhD
Senior Scientific Editor
EMBO Reports

--
